# LRRC8A is essential for hypotonicity-, but not for DAMP-induced NLRP3 inflammasome activation

Jack P Green[1,2†]*, Tessa Swanton[1,2†], Lucy V Morris[1,2], Lina Y El-Sharkawy[3], James Cook[1,2], Shi Yu[1,2], James Beswick[3], Antony D Adamson[4], Neil E Humphreys[4,5], Richard Bryce[3], Sally Freeman[3], Catherine Lawrence[1,2], David Brough[1,2]*

[1]Division of Neuroscience and Experimental Psychology, School of Biological Sciences, Faculty of Biology, Medicine and Health, Manchester Academic Health Science Centre, University of Manchester, Manchester, United Kingdom; [2]Lydia Becker Institute of Immunology and Inflammation, University of Manchester, Manchester, United Kingdom; [3]Division of Pharmacy and Optometry, School of Health Sciences, Faculty of Biology, Medicine and Health, Manchester Academic Health Science Centre, The University of Manchester, Manchester, United Kingdom; [4]Genome Editing Unit Core Facility, Faculty of Biology, Medicine and Health, University of Manchester, Manchester, United Kingdom; [5]EMBL-ROME, Epigenetics and Neurobiology Unit, Adriano Buzzati-Traverso Campus, Monterotondo, Italy

*For correspondence:
Jack.green@manchester.ac.uk
(JPG);
david.brough@manchester.ac.uk
(DB)

†These authors contributed
equally to this work

Competing interests: The
authors declare that no
competing interests exist.

Reviewing editor: Russell E
Vance, University of California,
Berkeley, United States

**Abstract** The NLRP3 inflammasome is a multi-molecular protein complex that converts inactive cytokine precursors into active forms of IL-1β and IL-18. The NLRP3 inflammasome is frequently associated with the damaging inflammation of non-communicable disease states and is considered an attractive therapeutic target. However, there is much regarding the mechanism of NLRP3 activation that remains unknown. Chloride efflux is suggested as an important step in NLRP3 activation, but which chloride channels are involved is still unknown. We used chemical, biochemical, and genetic approaches to establish the importance of chloride channels in the regulation of NLRP3 in murine macrophages. Specifically, we identify LRRC8A, an essential component of volume-regulated anion channels (VRAC), as a vital regulator of hypotonicity-induced, but not DAMP-induced, NLRP3 inflammasome activation. Although LRRC8A was dispensable for canonical DAMP-dependent NLRP3 activation, this was still sensitive to chloride channel inhibitors, suggesting there are additional and specific chloride sensing and regulating mechanisms controlling NLRP3.

## Introduction

Inflammation is an important protective host-response to infection and injury, and yet is also detrimental during non-communicable diseases (*Dinarello et al., 2012*). Inflammasomes are at the heart of inflammatory responses. Inflammasomes are formed by a soluble pattern recognition receptor (PRR), in many cases the adaptor protein ASC (apoptosis-associated speck-like protein containing a CARD), and the protease caspase-1 (2). Inflammasomes form in macrophages in response to a specific stimulus to drive the activation of caspase-1, facilitating the processing of the cytokines pro-interleukin (IL)1β and pro-IL-18 to mature secreted forms, and the cleavage of gasdermin D to cause pyroptotic cell death (*Evavold and Kagan, 2019*). A number of different inflammasomes have been described, but potentially the inflammasome of greatest interest to non-communicable disease is

**eLife digest** Inflammation is a critical part of a healthy immune system, which protects us against harmful pathogens (such as bacteria or viruses) and works to restore damaged tissues. In the immune cells of our body, the inflammatory process can be activated through a group of inflammatory proteins that together are known as the NLRP3 inflammasome complex.

While inflammation is a powerful mechanism that protects the human body, persistent or uncontrolled inflammation can cause serious, long-term damage. The inappropriate activation of the NLRP3 inflammasome has been implicated in several diseases, including Alzheimer's disease, heart disease, and diabetes. The NLRP3 inflammasome can be activated by different stimuli, including changes in cell volume and exposure to either molecules produced by damaged cells or toxins from bacteria. However, the precise mechanism through which the NLRP3 becomes activated in response to these stimuli was not clear.

The exit of chloride ions from immune cells is known to activate the NLRP3 inflammasome. Chloride ions exit the cell through proteins called anion channels, including volume-regulated anion channels (VRACs), which respond to changes in cell volume. Green et al. have found that, in immune cells from mice grown in the lab called macrophages, VRACs are the only chloride channels involved in activating the NLRP3 inflammasome when the cell's volume changes. However, when the macrophages are exposed to molecules produced by damaged cells or toxins from bacteria, Green et al. discovered that other previously unidentified chloride channels are involved in activating the NLRP3 inflammasome.

These results suggest that it might be possible to develop drugs to prevent the activation of the NLRP3 inflammasome that selectively target specific sets of chloride channels depending on which stimuli are causing the inflammation. Such a selective approach would minimise the side effects associated with drugs that generically suppress all NLRP3 activity by directly binding to NLRP3 itself. Ultimately, this may help guide the development of new, targeted anti-inflammatory drugs that can help treat the symptoms of a variety of diseases in humans.

formed by NLRP3 (NACHT, LRR and PYD domains-containing protein 3) (*Mangan et al., 2018*). The mechanisms of NLRP3 activation remain poorly understood.

The NLRP3 inflammasome is activated through several routes which have been termed the canonical, non-canonical, and the alternative pathways (*Mangan et al., 2018*). Activation of the canonical NLRP3 pathway, which has received greatest attention thus far, typically requires two stimuli; an initial priming step involving a pathogen associated molecular pattern (PAMP), typically bacterial endotoxin (lipopolysaccharide, LPS), to induce expression of pro-IL-1β and NLRP3, and a second activation step usually involving a damage associated molecular pattern (DAMP), such as adenosine triphosphate (ATP) (*Mariathasan et al., 2006*). In 1996, Perregaux and colleagues discovered that hypotonic shock was effective at inducing the release of mature IL-1β when applied to LPS-treated human monocytes and suggested the importance of a volume-regulated response (*Perregaux et al., 1996*). It was later discovered that hypotonicity induced release of IL-1β via activation of the NLRP3 inflammasome (*Compan et al., 2012*), and that this was linked to the regulatory volume decrease (RVD), which is a regulated reduction in cell volume in response to hypo-osmotic-induced cell swelling, and was inhibited by the chloride (Cl$^-$) channel blocker NPPB (5-nitro-(3-phenylpropylamino)benzoic acid) (*Compan et al., 2012*). The RVD is regulated by the Cl$^-$ channel VRAC (volume regulated anion channel). The molecular composition of the VRAC channel was established to consist of an essential LRRC8A sub-unit in combination with other (B-E) LRRC8 sub-units (*Qiu et al., 2014*; *Voss et al., 2014*). We recently reported that fenamate NSAIDs could inhibit the canonical NLRP3 inflammasome by blocking a Cl$^-$ channel, which we suggested could be VRAC (*Daniels et al., 2016*). We also further characterised the importance of Cl$^-$ flux in the regulation of NLRP3, showing that Cl$^-$ efflux facilitated NLRP3-dependent ASC oligomerisation (*Green et al., 2018*). Given the poor specificity of many Cl$^-$ channel inhibitors, we set out to systematically determine the importance of VRAC and the RVD to NLRP3 inflammasome activation. Given that hypertonic buffers inhibit ATP and other NLRP3 activating stimuli-induced IL-1β release from macrophages (*Perregaux et al., 1996*; *Compan et al., 2012*), we hypothesised that cell swelling,

and the RVD, would be central to all NLRP3 activating stimuli. Using pharmacological and genetic approaches, we discovered that VRAC exclusively regulated RVD-dependent NLRP3 activation in response to hypotonicity, and not NLRP3 activation in response to other canonical stimuli. Thus, we provide genetic evidence for the importance of Cl⁻ in regulating NLRP3 via the VRAC dependence of the hypotonicity response, and suggest the presence of additional Cl⁻ sensing mechanisms regulating NLRP3 in response to DAMPs.

## Results

Following publication of the cryo-electron microscopy (cryo-EM) structure of VRAC (*Deneka et al., 2018*) with inhibitor 4-(2-butyl-6,7-dichloro-2-cyclopentyl-indan-1-on-5-yl)oxobutyric acid (DCPIB) (*Figure 1A*; *Kern et al., 2019*), we were able to investigate the interaction of established VRAC inhibitors with the channel using molecular modelling. DCPIB was first computationally redocked using Molecular Operating Environment (MOE 2015.08, Chemical Computing Group, Canada) into the homohexameric VRAC structure (PDB code 6NZW, resolution 3.2 Å) (*Kern et al., 2019*). The resulting pose produced a reasonable overlay with the cryo-EM conformation of DCPIB, giving a root-mean-square deviation in atomic position of 2.6 Å (*Figure 1B,C*). DCPIB exhibits an ionic interaction of its carboxylate group with the cationic side-chain of one of the Arg103 residues comprising the electropositive selectivity filter of VRAC (*Kern et al., 2019*). Known VRAC inhibitors (*Daniels et al., 2016*; *Hélix et al., 2003*; *Droogmans et al., 1999*) possessing carboxylic acid groups (flufenamic acid (FFA)), mefenamic acid (MFA) and *N*-((4-methoxy)−2-naphthyl)−5-nitroanthranilic acid (MONNA), the tetrazole moiety (NS3728) and sulfonic acid groups (4-sulfonic calix[6]arene) were then docked into the DCPIB site of VRAC. The most favourably bound poses of these ligands were similarly found to block the pore in a cork-in-bottle manner (*Kern et al., 2019*) at the selectivity filter; the ligands' ionised acidic groups formed strong electrostatic interactions with Arg103 (*Figure 1D–H*). Tamoxifen, a basic inhibitor of VRAC was also docked into the cryo-EM structure (*Figure 1I*). Accordingly, tamoxifen docked with its cationic tertiary amino group remote to the Arg103 side-chains (*Figure 1I* and *Figure 1—figure supplement 1*); these side-chains instead formed cation-π interactions with the phenyl group of tamoxifen (*Figure 1I*). The interaction of the tamoxifen pose was computed as having a calculated ligand-binding affinity of −6.5 kcal mol⁻¹ via the molecular mechanics/generalised Born volume integration (MM/GBVI) method (*Labute, 2008*; *Galli et al., 2014*). The binding energies of the anionic ligands were also predicted as favourable, ranging from −5.1 (DCPIB) to −5.8 kcal mol⁻¹ (MONNA). This range excludes the larger 4-sulfonic calix[6]arene (calixarene), which gave a binding energy of −9.9 kcal mol⁻¹; we note that the GBVI implicit solvent model may be underestimating the high desolvation cost of this polyanionic ligand and therefore overestimating the magnitude of the corresponding binding energy of this compound.

We then tested the ability of six of the compounds described in *Figure 1* (DCPIB, calixarene (Calix), tamoxifen, FFA, MONNA, NS3728) to inhibit hypotonicity-induced VRAC-dependent Cl⁻ flux using the iodide (I⁻) quenching of halide-sensitive YFP H148Q/I152L (*Galietta et al., 2001*) in live HeLa cells (*Figure 2A*). In this model, I⁻ enters the cell through open Cl⁻ channels to induce quenching of a mutant EYFP. In response to hypotonic shock to induce VRAC opening, YFP fluorescence was immediately quenched, which was significantly inhibited by tamoxifen (10 μM), MONNA (50 μM), DCPIB (10 μM), FFA (100 μM), and NS3728 (10 μM), but not calixarene (100 μM) (*Figure 2A,B*). VRAC also regulates RVD in response to cell swelling (*Qiu et al., 2014*; *Voss et al., 2014*). We measured the RVD by measuring the change in cellular fluorescence in calcein-loaded primary mouse bone-marrow-derived macrophages (BMDMs) in response to hypotonicity. Hypotonicity caused a rapid increase in cell volume which declined over time, characteristic of an RVD response (*Figure 2C*). Similar to the quenching assay, RVD was also significantly inhibited in the presence of tamoxifen, MONNA, DCPIB, FFA and NS3728, but not calixarene (*Figure 2C,D*). These data suggest that all the molecules in our panel, except calixarene, are bona-fide VRAC inhibitors at the concentrations tested.

We tested whether the panel of VRAC inhibitors characterised above could block NLRP3 inflammasome activation and release of IL-1β in response to DAMP stimulation. Primary BMDMs were primed with LPS (1 μg mL⁻¹, 4 hr), before activation of NLRP3 by ATP (5 mM, 2 hr). Inhibitors were given at the same dose they inhibited VRAC above in HeLa cells 15 min before the addition of ATP

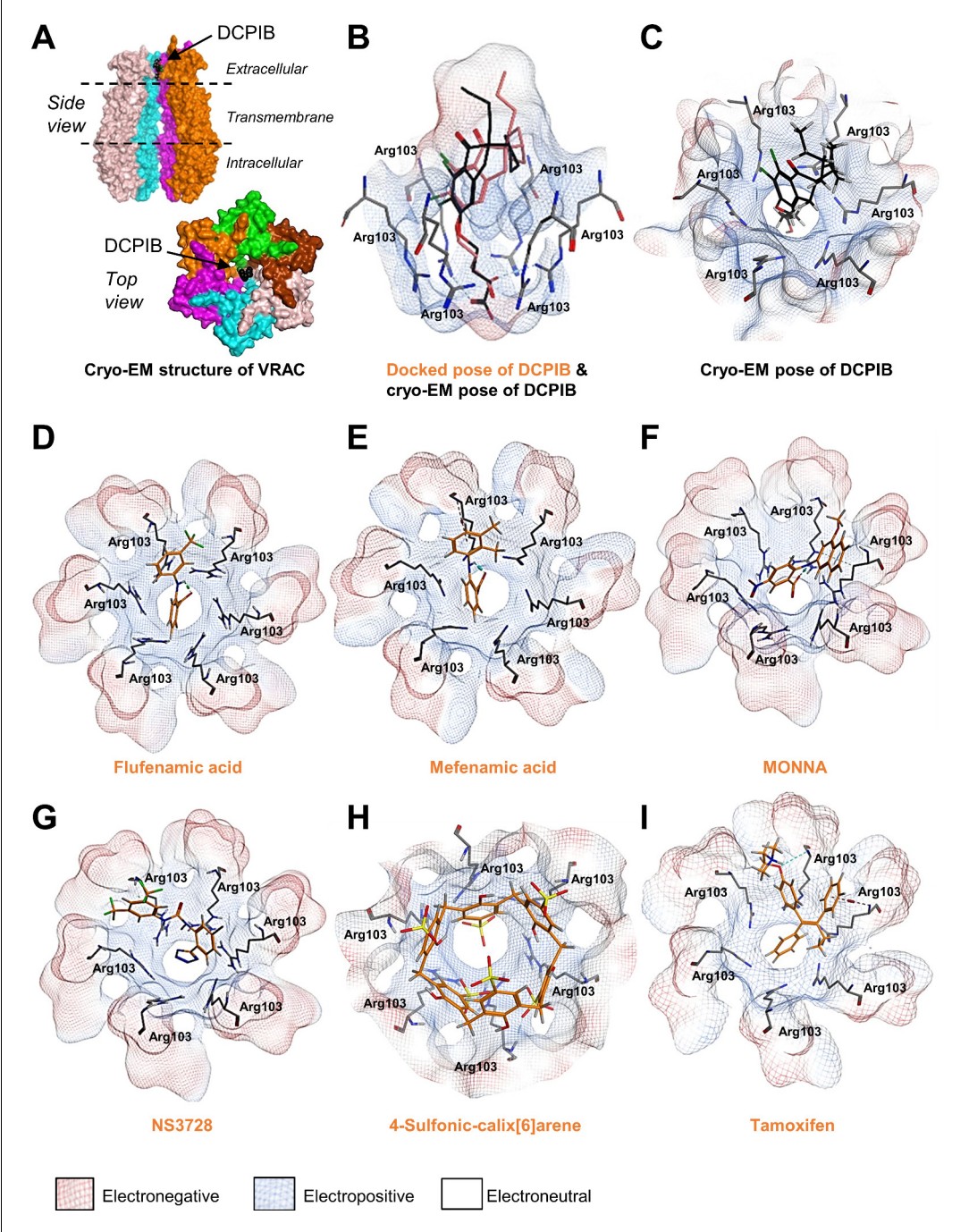

**Figure 1.** Modelling of proposed VRAC inhibitors on the Cryo-EM structure of LRRC8 channels. (**A**) Two orthogonal views: a side view with two chains removed from the hexameric LRRC8A protein with bound DCPIB and a top view for the hexameric VRAC channel (PDB:6NZW). (**B, D–I**) Docked VRAC inhibitors (orange) in the VRAC Arg103 extracellular selectivity filter. Protein surface coloured according to electrostatics: electronegative (red), electropositive (blue), and electroneutral (white). (**B**) MM/GBVI binding energies of docked DCPIB (orange) in VRAC ($-5.2$ kcal mol$^{-1}$) in a side view superimposed on the cryo-EM DCPIB pose (black). (**C**) Top-view of cryo-EM pose of DCPIB (black) in VRAC. (**D**) Flufenamic acid ($-5.1$ kcal mol$^{-1}$) (**E**) Mefenamic acid ($-5.2$ kcal mol$^{-1}$) (**F**) MONNA ($-5.8$ kcal mol$^{-1}$) (**G**) NS3728 ($-6.4$ kcal mol$^{-1}$) (**H**) 4-sulfonic calix[6]arene ($-9.9$ kcal mol$^{-1}$) (**I**) Tamoxifen ($-6.5$ kcal mol$^{-1}$).

The online version of this article includes the following figure supplement(s) for figure 1:

**Figure supplement 1.** Further modelling of VRAC inhibitors.

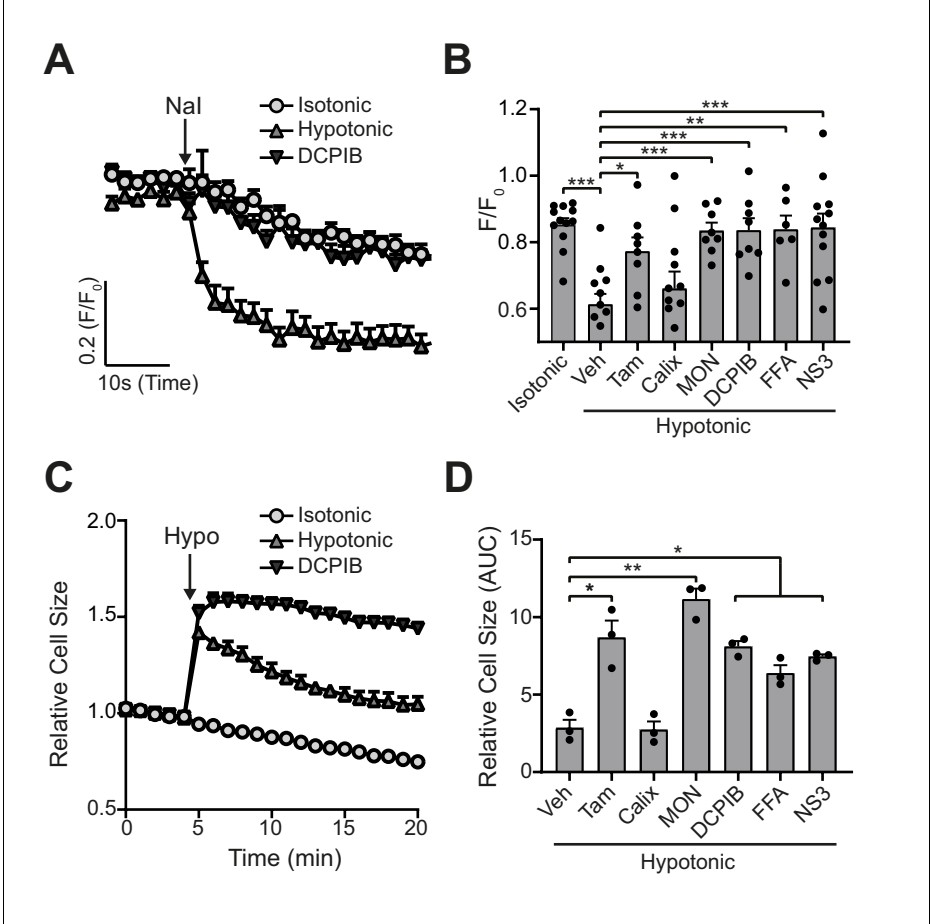

**Figure 2.** VRAC inhibitors block hypotonicity-induced Cl⁻ channel opening and regulatory volume decrease (RVD). (A) Cl⁻ channel opening measured in HeLa cells transiently expressing the halide-sensitive EYFP (H148Q/I152L). HeLa cells were pre-treated with a vehicle control (DMSO) or DCPIB (10 µM) and incubated in an isotonic (310 mOsm kg$^{-1}$) or hypotonic (215 mOsm kg$^{-1}$) solution for 5 min before quenching by addition of NaI (40 mM). (B) Normalised EYFP (H148Q/I152L) fluorescence values from HeLa cells pre-treated with either a vehicle control (DMSO), tamoxifen (Tam, 10 µM), 4-sulfonic calix[6]arene (Calix, 100 µM), MONNA (MON, 50 µM), DCPIB (10 µM), flufenamic acid (FFA, 100 µM) or NS3728 (NS3, 10 µM). Cells were incubated an isotonic (310 mOsm kg$^{-1}$) or hypotonic (215 mOsm kg$^{-1}$) solution for 5 min before quenching by addition of NaI. Fluorescent measurement was taken 30 s after NaI addition (n = 6–12). (C) Relative cell size of murine bone-marrow-derived macrophages (BMDMs) incubated in isotonic (340 mOsm kg$^{-1}$) or hypotonic (117 mOsm kg$^{-1}$) solution, pre-treated with a vehicle control (DMSO) or DCPIB (10 µM). BMDMs were labelled with the fluorescent dye calcein and area of fluorescence was measured over time. (D) Area under the curve of BMDMs incubated in a hypotonic solution (117 mOsm kg$^{-1}$) in the presence of either a vehicle control (DMSO), tamoxifen (Tam, 10 µM), 4-sulfonic calix[6]arene (Calix, 100 µM), MONNA (MON, 50 µM), DCPIB (10 µM), flufenamic acid (FFA, 100 µM) or NS3728 (NS3, 10 µM) (n = 3). *p<0.05, **p<0.01, ***p<0.01 determined by a one-way ANOVA with Dunnett's (vs vehicle control) post hoc analysis. Values shown are mean plus the SEM.

The online version of this article includes the following source data for figure 2:

**Source data 1.** Source data for data shown in *Figure 2A* and *Figure 2C*.

and were then present for the duration of the experiment. Of the panel of verified VRAC inhibitors, only MONNA, FFA and NS3728 consistently inhibited ATP-induced IL-1β release (*Figure 3A*). At the dose used in this assay, DCPIB did not consistently inhibit ATP-induced IL-1β release (*Figure 3A*), but at higher concentrations did inhibit NLRP3 activation (*Figure 3—figure supplement 1*). Likewise, pyroptosis, as measured by LDH release, was significantly reduced by MONNA and FFA, and was unaffected by tamoxifen or calixarene (*Figure 3B*). The inhibitors that blocked ATP-induced IL-1β release also inhibited ASC oligomerisation, caspase-1 activation, and gasdermin D cleavage

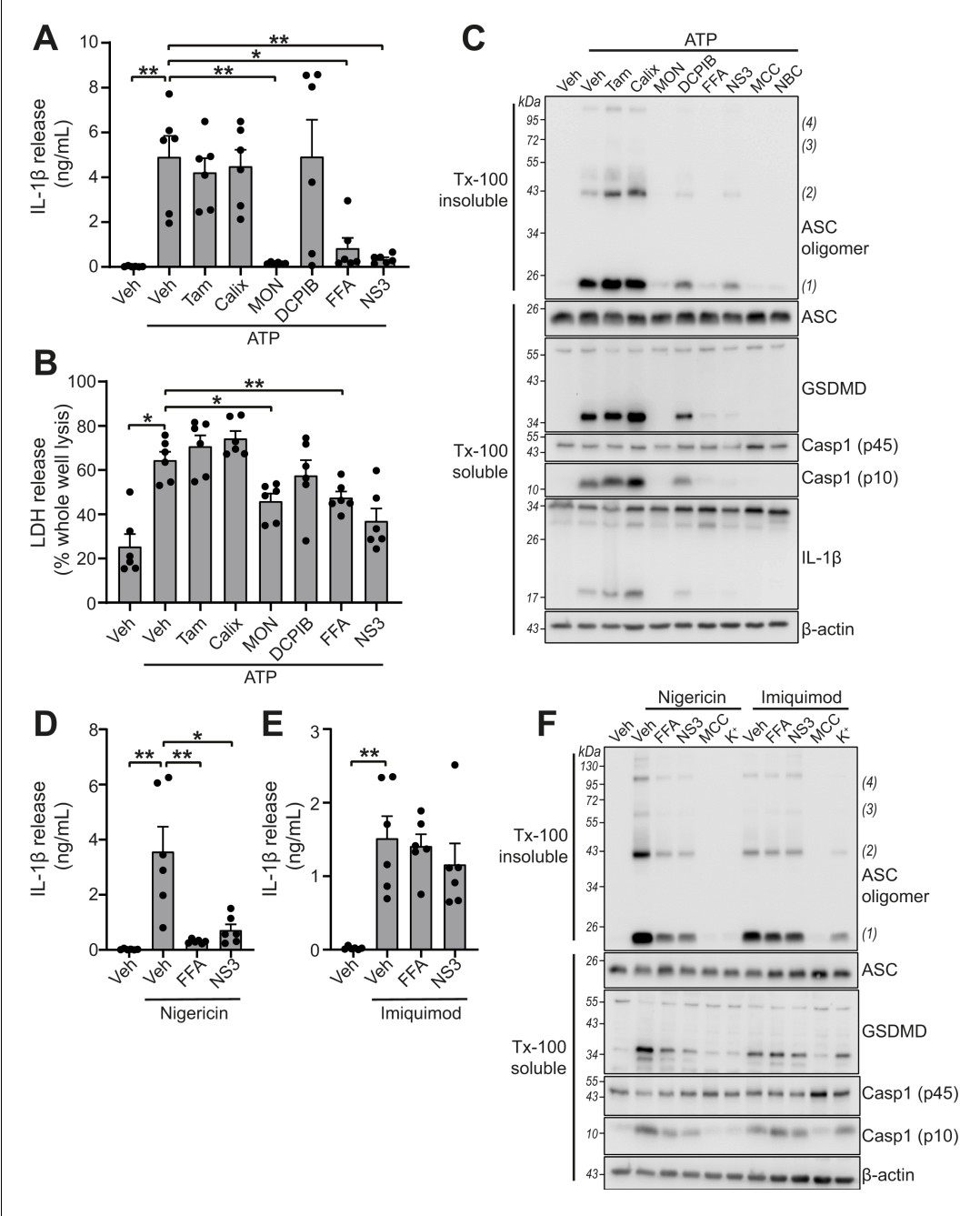

**Figure 3.** VRAC inhibitors differentially regulate NLRP3. (**A**) IL-1β release was determined by ELISA on supernatants from murine bone marrow derived macrophages (BMDMs). LPS-primed (1 μg mL$^{-1}$, 4 hr) BMDMs were pre-treated with either a vehicle control (DMSO), tamoxifen (Tam, 10 μM), 4-sulfonic calix[6]arene (Calix, 100 μM), MONNA (MON, 50 μM), DCPIB (10 μM), flufenamic acid (FFA, 100 μM), or NS3728 (NS3, 10 μM) before stimulation with ATP (5 mM, 2 hr) (n = 6). (**B**) Cell death determined by an LDH assay of cells treated in (**A**). (**C**) Western blot of Triton x-100 insoluble crosslinked ASC oligomers and soluble total BMDM cell lysates (cell lysate + supernatant) probed for ASC, GSDMD, caspase-1 and IL-1β. LPS-primed (1 μg mL$^{-1}$, 4 hr) BMDMs were pre-treated with either a vehicle control (DMSO), tamoxifen (Tam, 10 μM), 4-sulfonic calix[6]arene (Calix, 100 μM), MONNA (MON, 50 μM), DCPIB (10 μM), flufenamic acid (FFA, 100 μM), NS3728 (NS3, 10 μM) or the NLRP3 inhibitors MCC950 (MCC, 10 μM) and NBC19 (NBC, 10 μM) before stimulation with ATP (5 mM, 2 hr) (n = 3). (**D**) IL-1β release from LPS-primed (1 μg mL$^{-1}$, 4 hr) BMDMs pre-treated with a vehicle control (DMSO), flufenamic acid (FFA, 100 μM), or NS3728 (NS3, 10 μM) before stimulation with nigericin (10 μM, 2 hr) (n = 6). (**E**) IL-1β release from LPS-primed (1 μg mL$^{-1}$, 4 hr) BMDMs pre-treated with a vehicle control (DMSO), flufenamic acid (FFA, 100 μM) or NS3728 (NS3, 10 μM) before stimulation with imiquimod (75 μM, 2 hr) (n = 6). (**F**) Western blot of Triton x-100 insoluble crosslinked ASC oligomers and soluble total BMDM cell lysates (cell lysate + supernatant) probed for ASC, GSDMD, and caspase-1. LPS-primed (1 μg mL$^{-1}$, 4 hr) BMDMs were pre-treated with either a vehicle control (DMSO), flufenamic acid (FFA, 100 μM), NS3728 (NS3, 10 μM), the NLRP3 inhibitor MCC950 (MCC, 10 μM) or KCl (K$^+$, 25 mM) before stimulation with nigericin

*Figure 3 continued on next page*

*Figure 3 continued*

(10 μM, 2 hr) or imiquimod (75 μM, 2 hr) (n = 3). *p<0.05, **p<0.01, determined by a one-way ANOVA with Dunnett's (vs vehicle control) post hoc analysis. Values shown are mean plus the SEM.

The online version of this article includes the following figure supplement(s) for figure 3:

**Figure supplement 1.** Dose-response curve for DCPIB.

(*Figure 3C*). These data show that some very effective VRAC inhibitors failed to inhibit activation of the NLRP3 inflammasome and release of IL-1β, suggesting that VRAC may not be the molecular target of these molecules inhibiting the inflammasome.

ATP-induced NLRP3 activation is dependent upon $K^+$ efflux, whereas NLRP3 activation by treatment with the imidazoquinoline compound imiquimod is $K^+$ efflux-independent (*Groß et al., 2016*). We therefore sought to test if VRAC inhibitors that were effective at blocking ATP-induced inflammasome activation were specific to $K^+$ efflux-sensitive mechanisms. FFA (100 μM) and NS3728 (10 μM) were effective at blocking IL-1β release after treatment with the $K^+$ ionophore nigericin (10 μM, 2 hr) (*Figure 3D*), but were unable to block imiquimod (75 μM, 2 hr)-induced IL-1β release (*Figure 3E*). Similarly, ASC oligomerisation, caspase-1 activation and gasdermin D cleavage induced by nigericin were sensitive to FFA and NS3728 pre-treatment (*Figure 3F*). However, ASC oligomerisation, caspase-1 activation and gasdermin D cleavage induced by imiquimod were not affected by FFA and NS3728 pre-treatment (*Figure 3F*). Increased extracellular KCl (25 mM) was sufficient to block nigericin-induced activation, but not imiquimod, demonstrating the $K^+$ dependency of nigericin (*Figure 3F*). These data suggest that these $Cl^-$ channel inhibiting compounds exclusively target the $K^+$-dependent canonical pathway of NLRP3 activation.

Many $Cl^-$ channel inhibiting drugs are known to inhibit multiple $Cl^-$ channels, and we established that very effective VRAC inhibitors (tamoxifen and DCPIB) had negligible effect on NLRP3 activation at VRAC inhibiting concentrations. Thus, to conclusively determine the role of VRAC in NLRP3 inflammasome activation we generated a macrophage specific LRRC8A knockout (KO) mouse using CRISPR/Cas9 (*Figure 4A*). The generation of a macrophage-specific LRRC8A KO was required as whole animal LRRC8A KO mice do not survive beyond 4 weeks and have retarded growth (*Kumar et al., 2014*). *Lrrc8a*$^{fl/fl}$ mice were bred with mice constitutively expressing Cre under the *Cx3cr1* promoter, as previously shown to be expressed in monocyte and macrophage populations (*Yona et al., 2013*). This generated mice with the genotype *Lrrc8a*$^{fl/fl}$:*Cx3cr1*$^{cre}$ (KO) with littermates *Lrrc8a*$^{fl/fl}$:*Cx3cr1*$^{WT}$ (WT). Cell lysates were prepared from BMDMs and peritoneal macrophages isolated from WT and KO mice and were western blotted for LRRC8A confirming that *Lrrc8a* KO cells were knocked out for LRRC8A (*Figure 4B*). Functional loss of LRRC8A was confirmed using the calcein RVD assay described above. BMDMs were subjected to a hypotonic shock and changes in calcein fluorescence measured over time. In WT cells, there was a characteristic RVD (*Figure 4C*). However, in *Lrrc8a* KO cells there was complete loss of the RVD response (*Figure 4C,D*). The absence of RVD was also strikingly evident by observation of the cells by phase contrast microscopy (*Figure 4E*, *Videos 1* and *2*). Treatment of *Lrrc8a* KO BMDMs with DCPIB and subsequent hypotonic shock resulted in a further dysregulation of cell volume compared to WT cells or KO cells treated with hypotonic shock alone, suggesting that additional $Cl^-$ channels act to constrain cell volume in the absence of a functional RVD (*Figure 4—figure supplement 1*). These data confirm functional KO of the VRAC channel in macrophages.

We next used the *Lrrc8a* KO macrophages to test the hypothesis that VRAC and the RVD were important for NLRP3 inflammasome activation and IL-1β release in response to DAMP stimulation. WT BMDMs and *Lrrc8a* KO BMDMs were primed with LPS (1 μg mL$^{-1}$, 4 hr) and then treated with the NLRP3 inflammasome activators ATP (5 mM, 2 hr), nigericin (10 μM, 2 hr), silica (300 μg mL$^{-1}$, 2 hr), or imiquimod (75 μM, 2 hr). Knocking out LRRC8A had no effect on the release of IL-1β (*Figure 5A*) or cell death (*Figure 5B*). We then used western blotting to determine ASC oligomerisation and caspase-1 activation. In response to the NLRP3 inflammasome activators nigericin, ATP, and imiquimod, there was no effect of LRRC8A KO on ASC oligomerisation or caspase-1 activation (*Figure 5C*). Furthermore, IL-1β release in response to ATP or nigericin was still inhibited by the VRAC inhibitors flufenamic acid (FFA, 100 μM), and NS3728 (10 μM) in the *Lrrc8a* KO BMDMs, confirming that these inhibitors are inhibiting NLRP3 inflammasome activation by a VRAC-independent

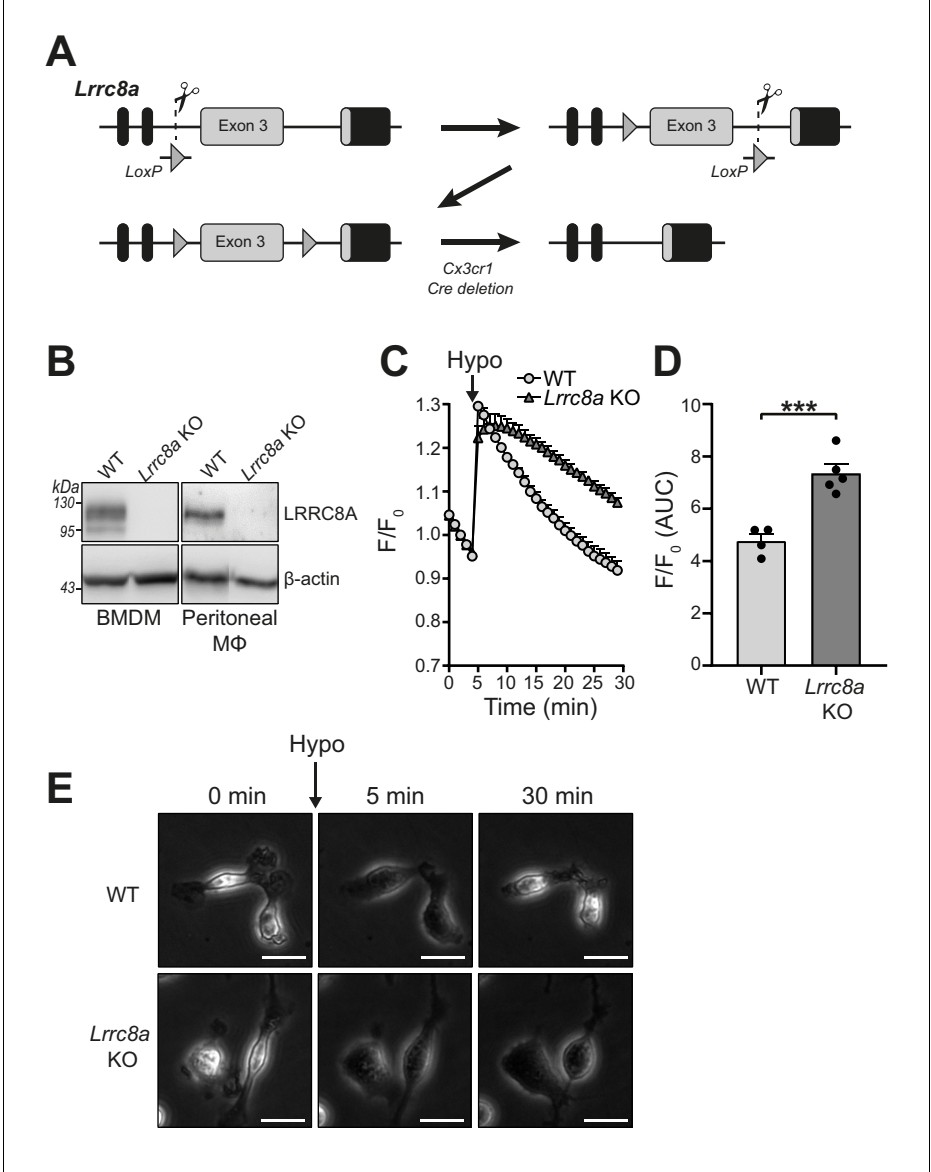

**Figure 4.** *Lrrc8a* KO macrophages are unable to undergo hypotonicity-induced regulatory volume decrease (RVD). (**A**) Generation of LRRC8A conditional allele. LRRC8A is found on mouse chromosome two and consists of four exons. Untranslated sequences are represented by black boxes, and coding sequences by grey boxes. Exon three contains the vast majority of coding sequence and was flanked by loxP sites in two sequential steps, first integrating the 5' LoxP by CRISPR-Cas9 (scissors) mediated double strand break and the supply of a homology flanked ssODN repair template containing the loxP site (grey triangle). This 5' fl background was then bred to establish a colony and the process repeated to integrate the second 3' loxP on this background. At each step, integration of loxP was confirmed by PCR and Sanger sequencing. Finally crossing with a Cre driver knocked into the *Cx3cr1* locus results in recombinase mediated excision of Exon 3. (**B**) Western blot of LRRC8A from wild-type (WT) or *Lrrc8a* knockout (KO) bone-marrow-derived macrophages (BMDMs) and peritoneal macrophages (Mφ) (n = 3). (**C**) Regulatory volume decrease measured by calcein fluorescence in WT or *Lrrc8a* KO BMDMs incubated in a hypotonic buffer (117 mOsm kg$^{-1}$) (n = 4–5). (**D**) Area under the curve (AUC) analysis of (**C**) (n = 4–5). (**E**) Representative phase contrast images of WT or *Lrrc8a* KO BMDMs incubated in a hypotonic buffer (117 mOsm kg$^{-1}$) at indicated time points (n = 3, Scale = 20 μm). ***p<0.001 determined by an unpaired *t*-test. Values shown are mean plus the SEM.

The online version of this article includes the following source data and figure supplement(s) for figure 4:

**Source data 1.** source data for data shown in *Figure 4C*.
**Figure supplement 1.** Effect of VRAC inhibitors on the RVD response of WT and *Lrrc8a* KO BMDMs.

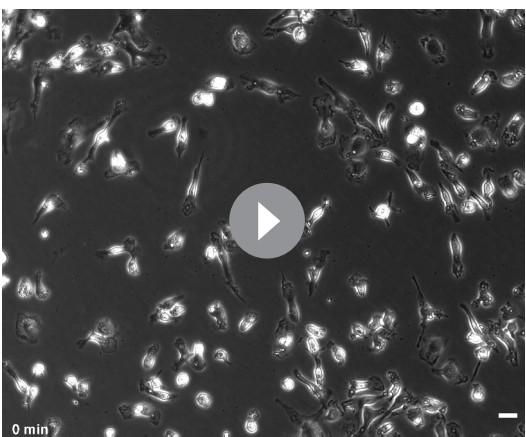

**Video 1.** Phase contrast time-lapse of the regulatory volume decrease of wild-type (WT) littermate bone marrow-derived macrophages (BMDMs). WT BMDMs were incubated in an isotonic buffer (340 mOsm kg$^{-1}$) for 5 min before dilution to a hypotonic solution (117 mOsm kg$^{-1}$) for the duration of the recording. Images were captured every minute (n = 3, Scale = 20 μm).
https://elifesciences.org/articles/59704#video1

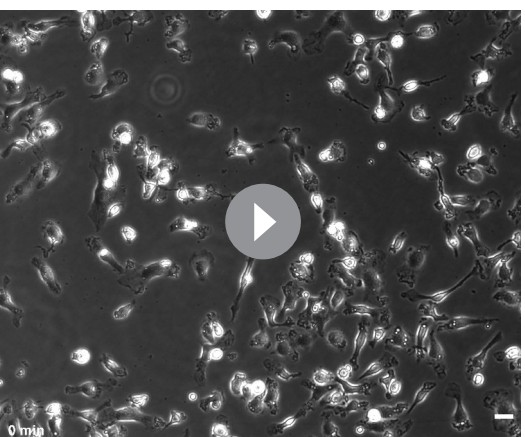

**Video 2.** Phase contrast time-lapse of the regulatory volume decrease of *Lrrc8a knockout* (KO) bone-marrow-derived macrophages (BMDMs). *Lrrc8a KO* BMDMs were incubated in an isotonic buffer (340 mOsm kg$^{-1}$) for 5 min before dilution to a hypotonic solution (117 mOsm kg$^{-1}$) for the duration of the recording. Images were captured every minute (n = 3, Scale = 20 μm).
https://elifesciences.org/articles/59704#video2

mechanism (*Figure 5D*). Flufenamic acid and NS3728 also inhibited ASC oligomerisation and caspase-1 activation as determined by western blot in the *Lrrc8a* KO BMDMs to the same extent as in the WT (*Figure 5E*). We then used a murine peritonitis model described previously (*Daniels et al., 2016*) to investigate the role of LRRC8A in vivo. First, we tested if the VRAC inhibitor NS3728 was effective at blocking NLRP3 in vivo. Wild-type C57BL6/J mice were injected intraperitoneally with NS3728 (50 mg kg$^{-1}$), the NLRP3 inhibitor MCC950 (50 mg kg$^{-1}$), or vehicle control, at the same time as LPS (1 μg, 4 hr). NLRP3 was then further activated by intraperitoneal injection of ATP (100 mM, 500 μL, 15 min) and IL-1β release was measured by ELISA of the peritoneal lavage (*Figure 5F*) and plasma (*Figure 5G*). Addition of ATP induced a significant increase in the release of IL-1β into peritoneal lavage and this was inhibited by MCC950 and NS3728, indicating an NLRP3-dependent response. IL-6 levels were unaltered in both peritoneal lavage and plasma by addition of NS3728 (*Figure 5—figure supplement 1A,B*). These data show that NS3728 was able to inhibit NLRP3 in vivo. To determine the role of VRAC in this model, we repeated this experiment in our macrophage *Lrrc8a* KO mice and their littermate controls. Macrophage *Lrrc8a* KO mice exhibited normal proportions of myeloid cells in the peritoneum as assessed by flow cytometry (*Figure 5—figure supplement 1C–G*). Loss of macrophage LRRC8A had no effect on the IL-1β levels in the peritoneal lavage in response to LPS and ATP (*Figure 5H*), or in the plasma (*Figure 5I*). Moreover, similar to our in vitro findings, NS3728 was still effective at inhibiting this response in the absence of LRRC8A (*Figure 5H,I*). These data suggested that VRAC was dispensable for NLRP3 activation by DAMP stimulation, and that the VRAC inhibitors are effective at inhibiting NLRP3 in the absence of VRAC, suggesting the presence of another target.

RVD in response to hypo-osmotic-induced cell swelling is documented as an inducer of NLRP3 inflammasome activation and IL-1β release (*Perregaux et al., 1996*; *Compan et al., 2012*). Since *Lrrc8a* KO BMDMs could no longer control their volume in response to hypotonic shock, we tested whether NLRP3 inflammasome activation by hypotonicity was altered. LPS-primed (1 μg mL$^{-1}$, 4 hr) BMDMs were incubated in a hypotonic solution (4 hr) which caused IL-1β and LDH release from WT cells, and which was significantly inhibited in *Lrrc8a* KO BMDMs (*Figure 6A,B*). There was no difference in IL-1β release or cell death between ATP-stimulated WT and KO BMDMs (*Figure 6A,B*). Caspase-1 cleavage and IL-1β processing induced by hypotonicity were also completely inhibited in the absence of LRRC8A (*Figure 6C*), indicating the response was completely dependent on both

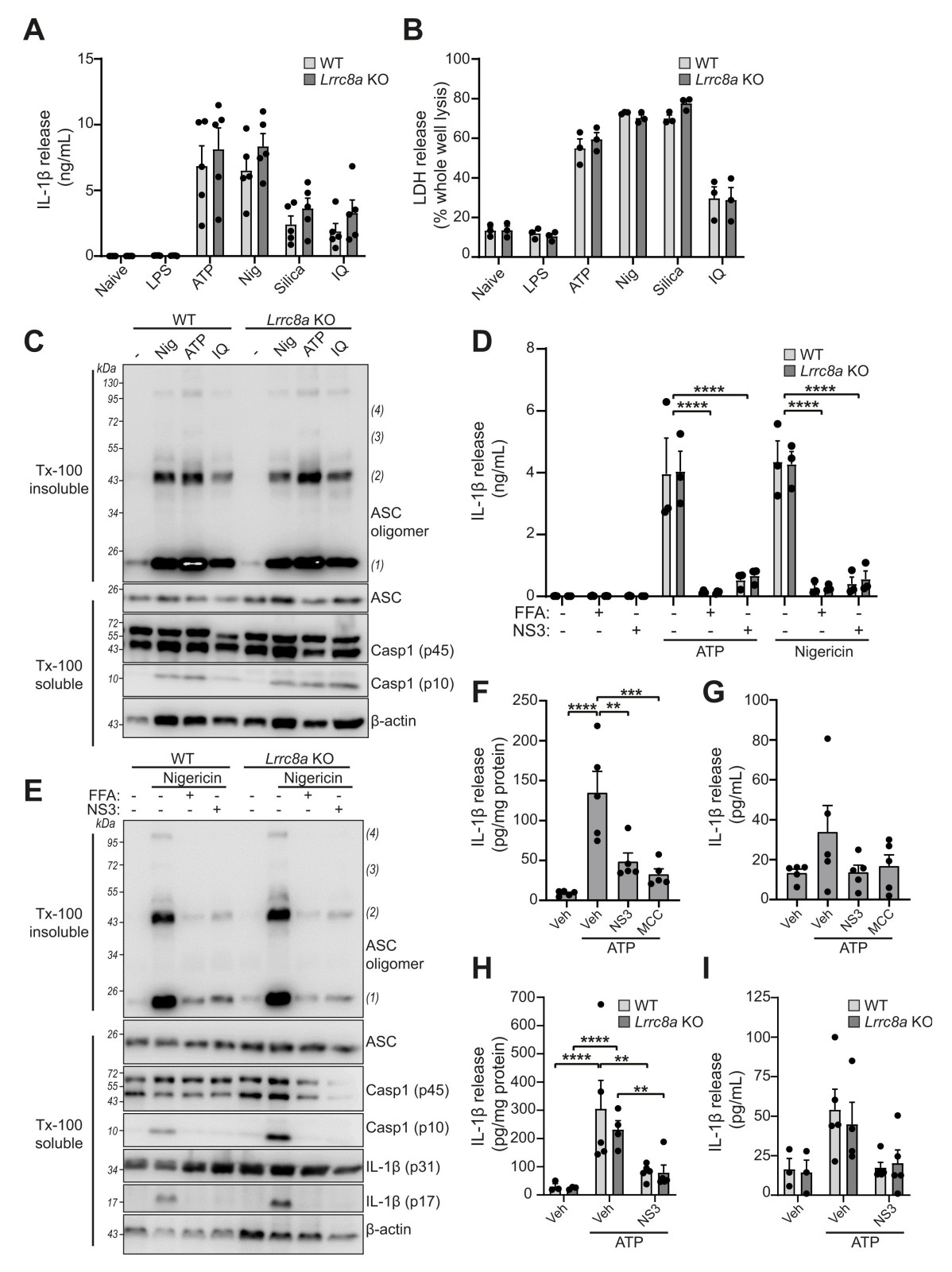

**Figure 5.** LRRC8A is dispensable for activation of the NLRP3 inflammasome. (**A**) IL-1β release was determined by ELISA on supernatants from wild-type (WT) or *Lrrc8a* knockout (KO) bone-marrow-derived macrophages (BMDMs). Naïve or LPS-primed (1 µg mL$^{-1}$, 4 hr) BMDMs were stimulated with either vehicle, ATP (5 mM), nigericin (Nig, 10 µM), silica (300 µg mL$^{-1}$) or imiquimod (IQ, 75 µM) for 2 hr (n = 5). (**B**) Cell death determined by an LDH assay of cells treated in (**A**) (n = 3). (**C**) Western blot of Triton x-100 insoluble crosslinked ASC oligomers and soluble total BMDM cell lysates (cell lysate +

*Figure 5 continued on next page*

Figure 5 continued

supernatant) probed for ASC and caspase-1. LPS-primed (1 µg mL$^{-1}$, 4 hr) WT or *Lrrc8a* KO BMDMs were stimulated with either nigericin (Nig, 10 µM), ATP (5 mM) or imiquimod (IQ, 75 µM) for 2 hr (n = 3). (D) IL-1β release from LPS-primed (1 µg mL$^{-1}$, 4 hr) WT or *Lrrc8a* KO BMDMs pre-treated with a vehicle control (DMSO), flufenamic acid (FFA, 100 µM) or NS3728 (NS3, 10 µM) and then stimulated with ATP (5 mM) or nigericin (10 µM) for 2 hr (n = 3). (E) Western blot of Triton x-100 insoluble crosslinked ASC oligomers and soluble total BMDM cell lysates (cell lysate + supernatant) probed for ASC, caspase-1 and IL-1β. LPS-primed (1 µg mL$^{-1}$, 4 hr) WT or *Lrrc8a* KO BMDMs were pre-treated with a vehicle control, flufenamic acid (FFA, 100 µM) or NS3728 (NS3, 10 µM) and stimulated with nigericin (10 µM, 2 hr) (n = 5). (F–G) IL-1β detected by ELISA in the peritoneal lavage (F) or plasma (G) from WT mice. Mice were pre-treated intraperitoneally (i.p.) with a vehicle control, NS3728 (NS3, 50 mg kg$^{-1}$) or MCC950 (MCC, 50 mg kg$^{-1}$) and LPS (1 µg). 4 hr after injection with LPS, mice were anaesthetised and injected with additional vehicle control, NS3728 (NS3, 50 mg kg$^{-1}$) or MCC950 (MCC, 50 mg kg$^{-1}$) before i.p. injection of ATP (100 mM, 500 µL, 15 min) (n = 5). (H–I) IL-1β detected by ELISA in the peritoneal lavage (H) or plasma (I) from *Lrrc8a* KO and WT littermates as treated in (F) (n = 3–5). **p<0.01, ***p<0.001, ****p<0.0001 determined by a one-way ANOVA with Dunnett's (vs vehicle control) post hoc analysis (F,G) or a two-way ANOVA with Tukey's post hoc analysis (A,B,D,H,I). Values shown are mean plus the SEM.

The online version of this article includes the following figure supplement(s) for figure 5:

**Figure supplement 1.** Loss of LRRC8a does not affect myeloid populations in the peritoneum.

NLRP3 and VRAC. Moreover, hypotonicity-induced ASC oligomerisation was also dependent on VRAC (*Figure 6D*). These data show that in response to hypo-osmotic stress, VRAC was essential for NLRP3 inflammasome activation.

## Discussion

Pharmacological and biochemical evidence supporting an important role of Cl$^-$ ions in the activation of the NLRP3 inflammasome has been provided by various studies over the years (*Perregaux et al., 1996*; *Compan et al., 2012*; *Daniels et al., 2016*; *Green et al., 2018*; *Verhoef et al., 2005*), although conclusive genetic evidence has been lacking. The promiscuous nature of many Cl$^-$ channel inhibiting drugs, and an unresolved molecular identity of major Cl$^-$ channels, have prevented the emergence of conclusive genetic proof. However, the discovery that the Cl$^-$ channel regulating the RVD (VRAC) was composed of LRRC8 sub-units, and that LRRC8A was essential for channel activity, offered us the opportunity to investigate the direct importance of VRAC in the regulation of NLRP3. Hypotonicity induces cell swelling which is corrected by the VRAC-dependent RVD (*Qiu et al., 2014*; *Voss et al., 2014*). The RVD was previously linked to NLRP3 activation (*Compan et al., 2012*). Thus by knocking out LRRC8A, and thus VRAC, we would discover that VRAC was essential for RVD-induced NLRP3 inflammasome activation, providing strong evidence for the direct requirement of a Cl$^-$ channel in NLRP3 inflammasome activation. Given that there are a number of inflammatory conditions and models that can be targeted by administration of a hyper-tonic solution (e.g. *Theobaldo et al., 2012*; *Schreibman et al., 2018*; *Shields et al., 2003*; *Petroni et al., 2015*), it is possible that the VRAC-dependent regulation of NLRP3 in response to hypotonicity could represent a therapeutic target.

However, VRAC was only essential for RVD-induced NLRP3 activation and was not involved in the NLRP3 response to DAMP stimulation. The fact that our VRAC channel-inhibiting drugs block DAMP-induced NLRP3 activation suggests that additional Cl$^-$ channels (or alternative targets) are involved in coordinating NLRP3 responses to other stimuli. Chloride intracellular channel proteins (CLICs 1–6) form anion channels and regulate a variety of cellular processes (*Littler et al., 2010*; *Argenzio and Moolenaar, 2016*). Localisation of CLIC1 and 4 to membrane fractions in macrophages is increased by LPS stimulation, and RNAi knockdown of both CLIC1 and 4 impaired LPS and ATP-induced IL-1β release from macrophages (*Domingo-Fernández et al., 2017*). In addition to CLICs 1 and 4, CLIC5 is also implicated in NLRP3-dependent IL-1β release (*Tang et al., 2017*). Knockdown of CLICs 1, 4, and 5 inhibits NLRP3 inflammasome activation in response to the soluble agonists ATP and nigericin, and also the particulate DAMP monosodium urate crystals (*Tang et al., 2017*). Thus, it appears that multiple Cl$^-$ channels encode diverse signals arising from DAMP stimulation, or from altered cellular homeostasis, to trigger NLRP3 inflammasome activation. Importantly our data suggest that Cl$^-$ channels are only important to NLRP3 activation dependent upon K$^+$ efflux, highlighting the further potential for selective pathway modulation and therapeutic development. The relationship between K$^+$ and Cl$^-$ efflux requires further investigation, and whether, in RVD, Cl$^-$ efflux is a pre-requisite for K$^+$ efflux.

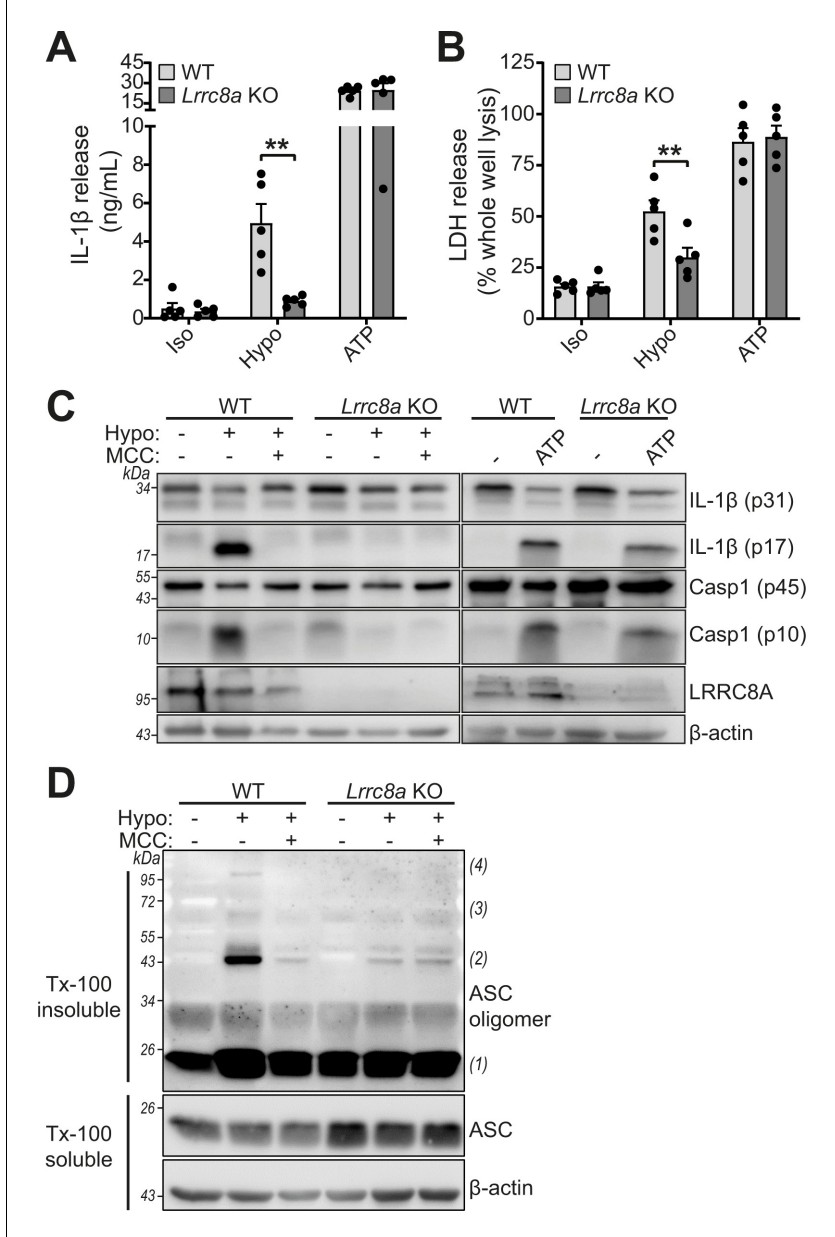

**Figure 6.** LRRC8A is an essential component of hypotonicity-induced NLRP3 activation. (**A**) IL-1β release was determined by ELISA on supernatants from wild-type (WT) or *Lrrc8a* knockout (KO) bone marrow derived macrophages (BMDMs). LPS-primed (1 μg mL$^{-1}$, 4 hr) wild-type (WT) or *Lrrc8a* knockout (KO) BMDMs were incubated in either an isotonic buffer (340 mOsm kg$^{-1}$), hypotonic buffer (117 mOsm kg$^{-1}$) or isotonic buffer with ATP (5 mM) for 4 hr (n = 5). (**B**) Cell death determined by an LDH assay of cells treated in (**A**). (**C**) Western blot of total BMDM cell lysates (cell lysate + supernatant) probed for IL-1β, caspase-1 and LRRC8A. LPS-primed (1 μg mL$^{-1}$, 4 hr) WT or *Lrrc8a* KO BMDMs were pre-treated with a vehicle control or a NLRP3 inhibitor MCC950 (MCC, 10 μM) before incubation in either an isotonic buffer (340 mOsm kg$^{-1}$), hypotonic buffer (117 mOsm kg$^{-1}$) or isotonic buffer with ATP (5 mM) for 4 hr. (**D**) Western blot of Triton x-100 insoluble crosslinked ASC oligomers and soluble total BMDM cell lysates (cell lysate + supernatant) probed for ASC. LPS-primed (1 μg mL$^{-1}$, 4 hr) WT or *Lrrc8a* KO BMDMs were pre-treated with a vehicle control or MCC950 (MCC, 10 μM) and incubated in either an isotonic buffer (340 mOsm kg$^{-1}$), hypotonic buffer (117 mOsm kg$^{-1}$) for 4 hr (n = 3). **p<0.01 determined by a two-way ANOVA with Tukey's post hoc analysis. Values shown are mean plus the SEM.

Although activation of VRAC is best understood under hypotonic conditions, VRAC activation has also been reported to occur in response to a variety of stimuli (*Osei-Owusu et al., 2018*). It is also possible that VRAC may regulate NLRP3 inflammasome activation in response to stimuli other than hypotonicity that were not tested here. For example, sphingosine-1-phosphate (S1P) activates VRAC in mouse macrophages (*Burow et al., 2015*), and we previously reported that sphingosine, and S1P, could activate NLRP3 (*Luheshi et al., 2012*). VRAC is also thought to be important for cell death induced by apoptosis inducing drugs (*Sørensen et al., 2016*; *Planells-Cases et al., 2015*), and we also previously reported that apoptosis inducing drugs could activate the NLRP3 inflammasome in activated macrophages (*England et al., 2014*) suggesting another potential area of relevance. While VRAC is directly permeable to Cl$^-$ it is also possible that additional Cl$^-$ channels are involved in the hypotonicity-induced swelling response. For example, VRAC activation caused by swelling can activate other Cl$^-$ channels, notably anoctamin 1 (e.g. *Liu et al., 2019*; *Benedetto et al., 2016*). Furthermore, whilst our research suggests an importance of Cl$^-$, VRAC is also permeable to small molecules including cGAMP (*Zhou et al., 2020*) and ATP (*Dunn et al., 2020*), highlighting additional ways through which VRAC could contribute to inflammation.

Inhibiting the NLRP3 inflammasome has become an area of intense research interest due to the multiple indications of its role in disease (*Mangan et al., 2018*). The inhibitor MCC950 is now thought to bind directly to NLRP3 to cause inhibition (*Coll et al., 2019*; *Tapia-Abellán et al., 2019*), although it has also been reported to inhibit Cl$^-$ flux from macrophages treated with nigericin (*Jiang et al., 2017*), and was found to bind directly to CLIC1 (*Laliberte et al., 2003*), so it is possible that some of its inhibitory activity may be attributable to an effect on Cl$^-$. We found that Cl$^-$ channel inhibition blocked IL-1β release in a NLRP3-dependent model of peritonitis, and previously reported protective effects of the fenamate NSAIDs in rodent models of Alzheimer's disease that we attributed to an effect on Cl$^-$ channel inhibition (*Daniels et al., 2016*). Thus, it is possible that targeting Cl$^-$ channels offers an additional route to inhibit NLRP3-dependent inflammation in disease.

In summary, we have reported that hypotonicity-induced NLRP3 inflammasome activation depends exclusively on the Cl$^-$ channel VRAC, and that different Cl$^-$ sensing and regulating systems coordinate the activation of NLRP3 in response to DAMPs. This opens the possibility of discrete Cl$^-$ regulating mechanisms conferring selectivity and information about the nature of the NLRP3 activating stimulus. Thus, this investigation has opened the door to further studies on Cl$^-$ regulation of NLRP3 and identified the possibility of selective therapeutic intervention strategies informed by the nature of the disease or DAMP stimulus, potentially minimising complications of immunosuppression caused by a blanket NLRP3 inhibition.

# Materials and methods

**Key resources table**

| Reagent type (species) or resource | Designation | Source or reference | Identifiers | Additional information |
|---|---|---|---|---|
| Genetic reagent (*Mus musculus*) | C57BL/6J | Charles River | C57BL/6NCrl | |
| Genetic reagent (*Mus musculus*) | C57BL/6J. LRRC8A[Em1Uman] (*Lrrc8a[fl/fl]*) | This paper | | Line maintained by David Brough lab, University of Manchester |
| Genetic reagent (*Mus musculus*) | B6J.B6N(Cg)-Cx3cr1tm1.1(cre)Jung/J | Jackson lab | Stock No: 025524 RRID:IMSR_JAX:025524 | Obtained from breeding colony managed by John Grainger lab (University of Manchester) |
| Cell line (*Mus musculus*) | Bone-marrow-derived macrophages (BMDMs) | In house | | Generated from bone marrow from above mouse lines |

*Continued on next page*

*Continued*

| Reagent type (species) or resource | Designation | Source or reference | Identifiers | Additional information |
|---|---|---|---|---|
| Cell line (*Mus musculus*) | Peritoneal macrophages | In house | | Generated by peritoneal lavage from above mouse lines |
| Cell line (*Homo Sapien*) | HeLa | ATCC | HeLa (ATCC CCL-2) RRID:CVCL_0030 | |
| Antibody | Anti-mouse IL-1β (goat polyclonal) | R and D Systems | AF-401-NA RRID:AB_416684 | (1:500) |
| Antibody | Anti-Caspase1 + p10 + p12 (rabbit monoclonal) | Abcam | ab179515 | (1:1000) |
| Antibody | Anti- mouse GSDMD (rabbit monoclonal) | Abcam | ab209845 RRID:AB_2783550 | (1:1000) |
| Antibody | Anti-mouse ASC/TMS1 (D2W8U) (rabbit monoclonal) | Cell Signaling Technology | 67824 RRID:AB_2799736 | (1:1000) |
| Antibody | LRRC8A (8H9) (mouse monoclonal) | Santa Cruz | sc-517113 | (1:200) |
| Antibody | Anti-β-Actin− Peroxidase (mouse monoclonal) | Sigma | A3854 RRID:AB_262011 | (1:20000) |
| Antibody | Anti-Rabbit Immunoglobulins HRP (goat polyclonal) | Agilent | P044801-2 RRID:AB_2617138 | (1:1000) |
| Antibody | Anti-Mouse Immunoglobulins HRP (rabbit polyclonal) | Agilent | P026002-2 RRID:AB_2636929 | (1:1000) |
| Antibody | Anti-Goat Immunoglobulins HRP (rabbit polyclonal) | Agilent | P044901-2 | (1:1000) |
| Recombinant DNA reagent | pcDNA3.1 Hygro EYFP H148Q/I152L | Addgene | 25874 RRID:Addgene_25874 | A gift from Peter Haggie |
| Commercial assay or kit | CytoTox 96 Non-Radioactive Cytotoxicity (LDH) Assay | Promega | G1780 | |
| Commercial assay or kit | Mouse IL-1β/IL-1F2 DuoSet ELISA | R and D systems | DY401 | |
| Chemical compound, drug | Lipopolysaccharides from *Escherichia coli* O26:B6 | Sigma | L2654 | For in vitro experiments |

*Continued on next page*

*Continued*

| Reagent type (species) or resource | Designation | Source or reference | Identifiers | Additional information |
|---|---|---|---|---|
| Chemical compound, drug | Lipopolysaccharides from *Escherichia coli* O127:B8 (in vivo) | Sigma | L3880 | For in vivo experiments |
| Chemical compound, drug | Adenosine Triphosphate (ATP) | Sigma | A2383 | |
| Chemical compound, drug | Nigericin sodium salt | Sigma | N7143 | |
| Chemical compound, drug | Silica | U.S. Silica | MIN-U-SIL 15 | |
| Chemical compound, drug | Imiquimod | InvivoGen | R837 | |
| Chemical compound, drug | Tamoxifen | Sigma | T5648 | |
| Chemical compound, drug | 4-Sulfonic calix[6]arene Hydrate | Thermo Fisher | 10494735 | |
| Chemical compound, drug | 4-[(2-Butyl-6,7-dichloro-2-cyclopentyl-2,3-dihydro-1-oxo-1H-inden-5-yl)oxy] butanoic acid (DCPIB) | Tocris | 1540 | |
| Chemical compound, drug | Flufenamic acid (FFA) | Sigma | F9005 | |
| Chemical compound, drug | NS3728 | David Brough Lab, University of Manchester | | |
| Chemical compound, drug | CP-456773 sodium salt (MCC950) | Sigma | PZ0280 | |
| Chemical compound, drug | NBC19 | David Brough Lab, University of Manchester | PMID:28943355 | |
| Chemical compound, drug | Calcein, AM, cell-permeant dye | Thermo Fisher | C1430 | |
| Chemical compound, drug | Disuccinimidyl suberate (DSS) | Thermo Fisher | 21555 | |

## Computational chemistry

Docking of ligands to VRAC employed the recently solved cryo-EM structure of VRAC (PDB: 6NZW, resolution 3.2 Å) (*Kern et al., 2019*). Tautomeric and ionisation states of VRAC amino acid residues at pH 7.4 were assigned using MOE (MOE 2015.08, Chemical Computing Group, Canada). Similarly, ligands were modelled in their ionised forms according to physiological conditions. Docking was performed with the Triangle Matcher placement method of MOE using the London dG scoring function (MOE 2015.08, Chemical Computing Group, Canada). The pocket into which the VRAC inhibitors were docked was that occupied by the (S)-isomer of DCPIB in the cryo-EM structure of VRAC. Rescoring of poses used the molecular mechanics (MM)/generalised Born/volume integral (GBVI) potential (*Labute, 2008*).

## Cell culture

Primary bone-marrow-derived macrophages (BMDMs) and peritoneal macrophages were isolated from male and female wild-type C57BL6/J mice. Bone marrow was harvested from both femurs, red blood cells were lysed and resulting marrow cells were cultured in 70% DMEM (10% v/v FBS, 100 U/mL penicillin, 100 μg/mL streptomycin) supplemented with 30% L929 mouse fibroblast-conditioned media for 6–7 days. BMDMs were seeded out the day before at a density of $1 \times 10^6$ mL$^{-1}$ in DMEM (10% v/v FBS, 100 U mL$^{-1}$ penicillin, 100 μg mL$^{-1}$ streptomycin). Peritoneal macrophages were isolated by peritoneal lavage and seeded out overnight at a density of $1 \times 10^6$ mL$^{-1}$ in DMEM (10% v/v FBS, 100 U mL$^{-1}$ penicillin, 100 μg mL$^{-1}$ streptomycin). HeLa cells were seeded out at $0.1 \times 10^6$ mL$^{-1}$ in DMEM (10% v/v FBS, 100 U mL$^{-1}$ penicillin, 100 μg mL$^{-1}$ streptomycin). HeLa cells were obtained from ATCC (HeLa (ATCC CCL-2)) and are periodically tested for mycoplasma.

## Inflammasome activation assays

Primary BMDMs were primed with LPS (1 μg mL$^{-1}$, 4 hr) in DMEM (10% v/v FBS, 100 U mL$^{-1}$ penicillin, 100 μg mL$^{-1}$ streptomycin). After priming, the media was replaced with serum-free DMEM, or when specified an isotonic buffer (132 mM NaCl, 2.6 mM KCl, 1.4 mM KH$_2$PO$_4$, 0.5 mM MgCl$_2$, 0.9 mM CaCl$_2$, 20 mM HEPES, 5 mM NaHCO$_3$, 5 mM Glucose, pH 7.3, 340 mOsm/kg) or hypotonic buffer (27 mM NaCl, 0.54 mM KCl, 0.3 mM KH$_2$PO$_4$, 0.5 mM MgCl$_2$, 0.9 mM CaCl$_2$, 20 mM HEPES, 5 mM NaHCO$_3$, 5 mM Glucose, pH 7.3, 117 mOsm kg$^{-1}$). When used, VRAC inhibitors were added 15 min before stimulation of the NLRP3 inflammasome.

For analysis of IL-1β release and pyroptosis, cell supernatants were collected. IL-1β release was determined by ELISA (DuoSet, R and D Systems) according to the manufacturer's instructions. Cell death was assessed by lactate dehydrogenase (LDH) release using CytoTox 96 nonradioactive cytotoxicity assay (Promega) according to manufacturer's instructions. For western blotting, total cell lysates were made by directly adding protease inhibitor cocktail and Triton x-100 (1% v/v) to each well containing cells and supernatant.

## ASC oligomerisation assay

$1 \times 10^6$ primary BMDMs were seeded out overnight into 12-well plates. After LPS priming (1 μg mL$^{-1}$, 4 hr), cells were incubated in either serum-free DMEM, an isotonic or a hypotonic buffer (as described above) and stimulated as described. BMDMs were lysed directly in-well by addition of protease inhibitor cocktail and Triton x-100 (1% v/v) and lysed on ice. Total cell lysates were then spun at 6800x*g* for 20 min at 4°C to separate the lysate into Triton x-100 soluble and insoluble fractions. The Triton x-100 insoluble fraction (pellet) was then chemically crosslinked by addition of disuccinimidyl suberate (DSS, 2 mM, 30 min, RT) in PBS. Following crosslinking, the insoluble fraction was spun at 6800x*g* for 20 min and the resulting pellet was resuspended and boiled in 40 μL 1X Laemlli buffer. The Triton x-100 soluble fraction was concentrated by trichloroacetic acid (TCA) precipitation. Triton x-100 soluble lysate was mixed 1:1 with TCA (20% w/v) and spun at 14,000x*g* for 10 min at 4°C. The pellet was then washed in acetone, spun at 14,000x*g* for 10 min at 4°C, and resuspended in 2X Laemlli buffer.

## Western blotting

Cell lysates were separated by Tris-glycine SDS PAGE and transferred onto nitrocellulose or PVDF membranes using a semidry Trans-Blot Turbo system (Bio-Rad). Membranes were blocked (1 hr, RT) in milk (5% w/v) in PBS containing Tween 20 (0.1% v/v, PBS-T) before incubation (overnight, 4°C) with indicated primary antibodies in bovine serum albumin (5% w/v) in PBS-T. Membranes were washed three times for 5 min in PBS-T before incubation (1 hr, RT) with appropriate HRP-conjugated secondary antibodies (Dako). After a further three washes in PBS-T, membranes were incubated with Amersham ECL prime detection reagent (GE healthcare) and chemiluminescence was visualised using a G:Box Chemi XX6 (Syngene).

## Regulatory volume decrease (RVD) assay

$5 \times 10^4$ BMDMs were seeded out into black walled 96-well plates overnight. Cells were loaded with calcein (10 μM, 1 hr, 37°C) in an isotonic buffer (132 mM NaCl, 2.6 mM KCl, 1.4 mM KH$_2$PO$_4$, 0.5 mM MgCl$_2$, 0.9 mM CaCl$_2$, 20 mM HEPES, 5 mM NaHCO$_3$, 5 mM Glucose, pH 7.3, 340 mOsm

kg$^{-1}$). Following loading, BMDMs were washed three times with isotonic buffer before incubation with VRAC inhibitors or vehicle control at indicated concentrations for 5 min. GFP fluorescence was then imaged for a further 5 min before hypotonic shock was induced by a fivefold dilution with a hypotonic buffer (0.9 mM CaCl$_2$, 20 mM HEPES, 5 mM NaHCO$_3$, 5 mM Glucose, pH 7.3), resulting in a final osmolarity of 117 mOsm kg$^{-1}$. GFP fluorescence was measured on an Eclipse Ti inverted microscope (Nikon) and analysed using Image J software. Point visiting was used to allow multiple positions to be imaged within the same time-course and cells were maintained at 37°C and 5% CO$_2$. For experiments with VRAC inhibitors, combined treatment with hypotonicity and VRAC inhibitors resulted in some cells undergoing lytic cell death over the course of the experiment and loss of calcein fluorescence. Therefore, GFP fluorescence was used to identify the area of living cells.

## Iodide YFP quenching assay

HeLa cells were seeded at a density of 0.1 × 10$^6$ ml$^{-1}$ in black-walled, clear bottom 96-well plates (Greiner). Transient transfection with the halide-sensitive YFP mutant pcDNA3.1 EYFP H148Q/I152L, a gift from Peter Haggie (Addgene plasmid # 25872), was performed using Lipofectamine 3000 (Thermo Fisher). 18–24 hr post-transfection, HeLa cells were washed twice with isotonic buffer (140 mM NaCl, 5 mM KCl, 20 mM HEPES, pH 7.4, 310 mOsm kg$^{-1}$) before 5 min incubation in 50 µL isotonic buffer containing either drug at indicated concentrations, or vehicle. 50 µL isotonic or hypotonic (5 mM KCl, 20 mM HEPES, 90 mM mannitol, pH 7.4, 120 mOsm kg$^{-1}$) buffer containing either drug or vehicle was then added and cells were incubated for a further 5 min. NaI (200 mM, 25 µL) was then added directly to the well, and fluorescence readings were take every 2 s using the FlexStation3 plate reader.

## Generation of *Lrrc8a*$^{fl/fl}$ mice

We used CRISPR-Cas9 to generate the floxed LRRC8A allele on C57BL/6J background. LRRC8A is a four exon gene spanning 26 kb on mouse chromosome 2. Only two of these exons contain coding sequence, with exon three harbouring >85% of the coding sequence and possessing large introns, and thus an ideal candidate for floxing. We initially attempted the 2-sgRNA, 2-oligo approach described previously (*Yang et al., 2013*), but failed to obtain mice with both loxP integrated on the same allele (*Gurumurthy et al., 2019*). Instead, a colony from a single founder with the 5' LoxP integrated was established, bred to homozygosity, and used as a background to integrate the second 3' loxP. For both steps, we used the Sanger WTSI website (http://www.sanger.ac.uk/htgt/wge/, *Hodgkins et al., 2015*) to design sgRNA that adhered to our criteria for off target predictions (guides with mismatch (MM) of 0, 1 or two for elsewhere in the genome were discounted, and MM3 were tolerated if predicted off targets were not exonic). sgRNA sequences were purchased as crRNA oligos, which were annealed with tracrRNA (both oligos supplied IDT; Coralville, USA) in sterile, RNase-free injection buffer (TrisHCl 1 mM, pH 7.5, EDTA 0.1 mM) by combining 2.5 µg crRNA with 5 µg tracrRNA and heating to 95°C, which was allowed to slowly cool to room temperature.

For 5' targeting the sgRNA GTCTAGTTAGGGACTCCTGG-*GGG* was used, with the ssODN repair template 5'-tccttgacttgctgtttaccgctctcttccccacaccacagttatccacaggaagttacccataacctccctcgtg-cacccctacccccaATAACTTCGTATAGCATACATTATACGAAGTTATGGTACCggagtccctaacta-gacctgctgtctctccatagccctgtctacacct-3', where capitals indicate the LoxP sequence with a KpnI site, and lower case the homology arms. For embryo microinjection, the annealed sgRNA was complexed with Cas9 protein (New England Biolabs) at room temperature for 10 min, before addition of ssODN (IDT) donor (final concentrations; sgRNA 20 ng µL$^{-1}$, Cas9 protein 20 ng µL$^{-1}$, ssODN 50 ng µL$^{-1}$). CRISPR reagents were directly microinjected into C57BL6/J (Envigo) zygote pronuclei using standard protocols (*Demayo et al., 2012*). Zygotes were cultured overnight and the resulting two-cell embryos surgically implanted into the oviduct of day 0.5 post-coitum pseudopregnant mice. Potential founder mice were identified by extraction of genomic DNA from ear clips, followed by PCR using primers that flank the homology arms and sgRNA sites (Geno F1 tcagatggcgaaccagaagtc and Geno R1 tacaatgtagtcaggtgtgacg). WT sequences produced a 833 bp band, and loxP knock in 873 bp, which is also susceptible to KpnI digest. Pups with a larger band were reserved, the band isolated and amplified using high fidelity Phusion polymerase (NEB), gel extracted and subcloned into pCRblunt (Invitrogen). Colonies were mini-prepped and Sanger sequenced with M13 Forward and Reverse primers, and aligned to predicted knock-in sequence. Positive pups were bred with a WT

C57BL6/J to confirm germline transmission and a colony established. To integrate the 3' LoxP we used sgRNA ACTACCCCATTACCTCTTGG-*TGG* with the ssODN repair template 5'- gagggc-caaaactgtgtggaaagcaacacccttgaagtgtaggtggcccctgtgcaccagctctgtgtgtgactgcaaagcccccaccaagaA TAACTTCGTATAGCATACATTATACGAAGTTATGGTACCggtaatggggtagttagacgggctgagggcagag-cacttgtgtggctt-3'. Again, capitals indicate the LoxP sequence with a KpnI site, and lower case the homology arms.

For this second round of targeting, we generated embryos from the LRRC8A-5'fl colony by IVF using homozygous LRRC8A-5'fl mice, and used electroporation (Nepa21 electroporator, Sonidel) to deliver the sgRNA:Cas9 RNP complex and ssODN to the embryos, AltR crRNA:tracrRNA:Cas9 complex (200 ng μL$^{-1}$; 200 ng μL$^{-1}$; 200 ng μL$^{-1}$ respectively) and ssDNA HDR template (500 ng μL$^{-1}$) (*Kaneko, 2017*). Zygotes were cultured overnight and the resulting two-cell embryos surgically implanted into the oviduct of day 0.5 post-coitum pseudopregnant mice. Potential founder mice were identified by extraction of genomic DNA from ear clips, followed by PCR using primers that flank the homology arms and sgRNA sites (Geno F2 atccccactgcttttctgga and Geno R2 ccactcaa-gagccagcaatg). WT sequences produced a 371 bp band, and loxP knock in 411 bp, which is also susceptible to KpnI digest. As before, Pups with a larger band were reserved, the band isolated and amplified using high fidelity Phusion polymerase (NEB), gel extracted and subcloned into pCRblunt (Invitrogen). Colonies were mini-prepped and Sanger sequenced with M13 Forward and Reverse primers, and aligned to predicted knock-in sequence. Positive pups were bred with a WT C57BL6/J to confirm germline transmission and a colony established (*Lrrc8a$^{fl/fl}$*). *Lrrc8a$^{fl/fl}$* mice were bred with *Cx3cr1$^{cre}$* mice (as previously described *Yona et al., 2013*) to generate *Lrrc8a$^{fl/fl}$ Cx3cr1$^{cre/+}$* mice, which specifically induce removal of LRRC8A in cells expressing CX3CR1 (*Lrrc8a KO*). CX3CR1-cre mice were obtained from a breeding colony at the University of Manchester managed by John Grainger. Experiments using *Lrrc8a* KO cells were compared to wild-type littermate controls (*Lrrc8a$^{fl/fl}$ Cx3cr1$^{+/+}$*).

## In vivo peritoneal inflammation model

All procedures were performed with appropriate personal and project licenses in place, in accordance with the Home Office (Animals) Scientific Procedures Act (1986), and approved by the Home Office and the local Animal Ethical Review Group, University of Manchester. Eight to ten week-old male wild type (WT) C57BL/6J mice (Charles River) were used to test efficacy of NS3728 and MCC950. Mice were treated intraperitoneally with LPS (2 μg mL$^{-1}$, 500 μL) and either a vehicle control (5% DMSO (v/v), 5% Cremophor (v/v), 5% ethanol (v/v) in PBS), NS3728 (50 mg kg$^{-1}$) or MCC950 (50 mg kg$^{-1}$) for 4 hr. Mice were then anesthetised with isofluorane (induced at 3% in 33% O$_2$, 67% NO$_2$, maintained at 1–2%) before injection with a vehicle control, NS3728 or MCC950 as before and ATP (0.5 mL, 100 mM in PBS, pH 7.4) or PBS for 15 min. The peritoneum was then lavaged with RPMI 1640 (3 mL) and blood was collected via cardiac puncture. For experiments using *Lrrc8a* knockout (KO) mice, 8- to 10-week-old male and female *Lrrc8a* KO and WT littermate controls were used and treated as described above. Plasma and lavage fluid was used for cytokine analysis. BCA analysis was performed on the peritoneal lavage fluid to normalise cytokine release to total protein level. Murine studies were performed with the researcher blinded to genotype and treatment for the duration of the experiment.

## Flow cytometry

Eight to ten-week-old male and female *Lrrc8a* KO and WT littermate controls were anesthetised with isofluorane (induced at 3% in 33% O$_2$, 67% NO$_2$, maintained at 1–2%) and the peritoneal cavity was lavaged with 6 mL RPMI (3% v/v FBS, 100 U mL$^{-1}$ penicillin, 100 μg mL$^{-1}$ streptomycin, 1 mM EDTA) before sacrifice. Red blood cells were lysed using Pharmlyse (BD Biosciences) in H$_2$O. Cells were surface stained with fluorescence-conjugated anti-CD45, anti-CD11b, anti-Ly6G, anti-MHCII, anti-F4/80, anti-Ly6C and anti-CX3CR1 antibody cocktail containing Fc block (anti-CD16/CD32) and Tris-EDTA (1 mM). Cells were then fixed (10 min, room temperature) with paraformaldehyde (2% w/v). Live/Dead Fixable Blue stain was used to exclude dead cells. Samples were analysed on an LSRII flow cytometer (Becton-Dickinson) and cell populations characterised as follows:- neutrophils (CD45$^{hi}$/CD11b$^{hi}$/Ly6G$^{hi}$), monocyte-derived-macrophages (CD45$^{hi}$/CD11b$^{hi}$/Ly6G$^-$/MHCII$^{hi}$/F4/80$^-$),

resident macrophages (CD45$^{hi}$/CD11b$^{hi}$/Ly6G$^-$/F4/80$^{hi}$), Ly6C$^{hi}$ monocytes (CD45$^{hi}$/CD11b$^{hi}$/Ly6G$^-$/MHCII$^-$/F4/80$^-$/CX3CR1$^{hi}$/Ly6C$^{hi}$), using FlowJo software.

## Quantification and statistical analysis

Data are presented as mean values plus the SEM. Accepted levels of significance were *$p<0.05$, **$p<0.01$, ***$p<0.001$, ****$p<0.0001$. Statistical analyses were carried out using GraphPad Prism (version 8). Data where comparisons were made against a vehicle control, a one way ANOVA was performed with a Dunnett's *post hoc* comparison was used. Experiments with two independent variables were analysed using a two-way ANOVA followed by a Tukey's *post hoc* corrected analysis. Equal variance and normality were assessed with Levene's test and the Shapiro–Wilk test, respectively, and appropriate transformations were applied when necessary. n represents experiments performed on individual animals or different passages for experiments involving HeLa cells.

## Acknowledgements

This work was supported by Medical Research Council Grants MR/N029992/1 and MR/T0116515/1 (to DB), The Alzheimer's Society AS-PhD-16-002 (to DB) and by a Presidential Fellowship (University of Manchester, to JG).

## Additional information

### Funding

| Funder | Grant reference number | Author |
| --- | --- | --- |
| Medical Research Council | MR/N029992/1 | David Brough |
| Medical Research Council | MR/T0116515/1 | David Brough |
| Alzheimer Society | AS-PhD-16-002 | David Brough |
| University of Manchester | Presidential Fellowship | Jack P Green |

The funders had no role in study design, data collection and interpretation, or the decision to submit the work for publication.

### Author contributions

Jack P Green, Data curation, Formal analysis, Investigation, Visualization, Methodology, Project administration, Writing - review and editing; Tessa Swanton, Data curation, Formal analysis, Investigation, Visualization, Methodology, Writing - review and editing; Lucy V Morris, Formal analysis, Investigation, Methodology, Writing - review and editing; Lina Y El-Sharkawy, Formal analysis, Investigation, Writing - review and editing; James Cook, Investigation, Writing - review and editing; Shi Yu, Investigation, Methodology, Writing - review and editing; James Beswick, Antony D Adamson, Neil E Humphreys, Resources, Investigation, Methodology, Writing - review and editing; Richard Bryce, Formal analysis, Supervision, Investigation, Methodology, Writing - review and editing; Sally Freeman, Resources, Formal analysis, Supervision, Methodology, Writing - review and editing; Catherine Lawrence, Supervision, Funding acquisition, Methodology, Project administration, Writing - review and editing; David Brough, Conceptualization, Resources, Supervision, Funding acquisition, Writing - original draft, Project administration, Writing - review and editing

### Author ORCIDs

Jack P Green (iD) https://orcid.org/0000-0003-1108-3422
Shi Yu (iD) http://orcid.org/0000-0003-0651-4769
David Brough (iD) https://orcid.org/0000-0002-2250-2381

### Ethics

Animal experimentation: All procedures were performed with appropriate personal licenses in place, under the project license P0E90B9E2, in accordance with the Home Office (Animals) Scientific

Procedures Act (1986), and approved by the Home Office and the local Animal Ethical Review Group, University of Manchester.

## Decision letter and Author response

Decision letter https://doi.org/10.7554/eLife.59704.sa1
Author response https://doi.org/10.7554/eLife.59704.sa2

## Additional files

### Supplementary files

• Transparent reporting form

### Data availability

All data generated or analysed during this study are included in the manuscript and supporting files. Source data are provided for graphs summarised where individual data points are not already provided in the figure, i.e Figure 2A, 2C and 4C.

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
