## [Decision Letter]

**Acceptance summary:**

This paper addresses how an important immune sensor (called the NLRP3 inflammasome) is regulated by chloride channels, particularly in response to hypotonic shock. A specific channel component is shown, using complementary genetic and pharmacological methods, to play a critical role in regulating the inflammatory response to hypotonic shock. The study thus helps provide a molecular basis for a longstanding inflammatory stimulus that may have relevance in diverse human diseases.

**Decision letter after peer review:**

Thank you for submitting your article "LRRC8A regulates hypotonicity-induced NLRP3 inflammasome activation" for consideration by *eLife*. Your article has been reviewed by Tadatsugu Taniguchi as the Senior Editor, a Reviewing Editor, and three reviewers. The following individuals involved in review of your submission have agreed to reveal their identity: Stephen Graf Brohawn (Reviewer #2).

The reviewers have discussed the reviews with one another and the Reviewing Editor has drafted this decision to help you prepare a revised submission.

In their manuscript "LRRC8A regulates hypotonicity-induced NLRP3 inflammasome activation" Green et al., investigate the role of VRACs in NLRP3 inflammasome activation. They confirm previously published observations (including from the author's group) that several VRAC (and other Cl^-^ channel) inhibitor can block NLRP3 inflammasome activation. Intriguingly, the authors report that the ability to block VRAC does not correlate with the ability to block NLRP3 inflammasome activation. They conclude that off target effects have to account for NLRP3 blocking activity by VRAC inhibitors. Next, the authors provide genetic tools to study the role of VRAC in NLRP3 inflammasome activation and find that canonical NLRP3 activators do not require VRAC. However, regulated volume decrease (RVD)-mediated NLRP3 activation in hypo-osmolar conditions does require VRAC. Overall, the consensus opinion of the reviewers is that the work addresses an important question and is overall well done. The following points are requested to be addressed in a revised version of the manuscript:

** Note that consistent with the *eLife* COVID revision policy, the below experimental suggestions are not required, though the reviewers feel they would improve the manuscript if adding data is possible. If adding data is not possible, then revisions to the text will suffice **

Points requiring additional experimentation:

1) The pharmacological analysis of VRAC inhibition is done entirely in Hela cells whereas the rest of the paper is about NLRP3 activation in macrophages.

a) Given that subunit composition could be different between HeLa cells and macrophages and this could play a role in differing pharmacology, it is imperative that the analysis of VRAC inhibition should be performed on macrophages, not the endogenous channels in Hela cells.

b) In addition, the authors should confirm that the VRAC KOs are completely deficient in RVD by using their VRAC inhibitors in VRAC KOs. In other words Figure 4C should include several of the VRAC inhibitors in both WT and VRAC KO cells.

2) The cell culture experiments showing hypotonicty-induced NLRP3 activation are convincing, but raise the question: under what circumstance(s) is this physiologically relevant?

a) There seems to be an underlying implication or assumption that DAMP stimulation evokes VRAC activation. Is this the intendend implication? If so, this needs to be explicitly tested.

b) Alternatively, the authors should provide another physiologically relevant scenario in which hypotonicity might naturally be encountered by a cell, leading to NLRP3 activation. Do the authors think that the extreme (though typical in the literature) hypotonic conditions used in culture experiments (using a switch from ~340to ~120 mOsm solutions) are likely to occur in vivo? Under what circumstances?

c) VRAC and LRRC8-contianing channels have been proposed to be activated by stimuli other than reduced intracellular ionic strength. Could these alternatives be involved in LRRC8-mediated NLRP3 activation?

d) Finally, can the authors speculate on whether it is Cl^-^ flux per se through LRRC8 or other Cl^-^ channels that is important for NLRP3 activation? Other possibilities include voltage changes, RVD, and flux of other permeant ions like ATP or cyclic dinucleotides.

e) In addition to the above (or as a result of addressing the above), the authors should expand on the discussion of potential physiological implications.

3) The analysis of VRAC channel function based solely on the YFP-quenching assay is technically inadequate. The differential effects on NLRP3 activation of the different inhibitors is a potentially important finding, but this claim cannot be made solely on this one indirect readout. The YFP quenching method is unlikely to have sufficient temporal resolution and dynamic range to detect smaller changes (or lack of changes) in ionic currents. Direct electrophysiological analysis of the chloride currents is required to test if there are differences in the mechanism or time course of inhibition that could explain the lack of effects on DAMP mediated inflammasome activation.

4) The authors should investigate if VRAC-mediated Cl^-^efflux enables K^+^ efflux to drive NLRP3 inflammasome activation during hypo-osmolar conditions. They should determine the intracellular K^+^ and Cl^-^ content in presence of pharmacological NLRP3 inhibition in WT vs VRAC KO cells. I think it'll be important to correlate the efflux of K^+^ with NLRP3 activation during RVD, especially in light of PMID: 22981536, which claims otherwise (using pharmacological inhibition).

Points requiring textual revisions to the manuscript:

1) The title should be reworked to include the key finding that LRRC8a does not regulate canonical NLRP3 inflammsome activation as this is a central part of the paper. The title only reflects Figure 6 while most of the work shows that LRRC8a does not regulate the canonical NLRP3 inflammasome.

2) There is an underlying assumption that the inhibitors that inhibit canonical (ATP for instance) NLRP3 activation (MON, FFA, NS3) likely inhibit another Cl^-^ channel. This is part of the discussion and interpretation of their data and it appears there is no rational basis for this conclusion. The mechanism for how these compounds block NLRP3 remains a mystery and could very likely include other unknown targets.

3) The authors are suggested to modify the presentation of Figure 1 for clarity.

a) In (A), two orthogonal views (a side view with one or two LRRC8A subunits in the front removed and a top view) and the addition of lines indicating membrane boundaries would give better perspective.

b) The legend states B-I are colored with "hydrophilic (blue), hydrophobic regions (red) and neutral regions (white)." It is recommended to change the coloring in B-I are colored to the more typical blue (electropositive) to white (electroneutral) to red (electronegative) as in Figure 1—figure supplement 1. Or to a green-purple hydrophobic/hydrophilic color scheme as red/white/blue is typically used for electronegative-electropositive coloring. In either case, please add a key to the figure with the color scale.

c) Consider adding labels for each drug in the panels.

4) While the docking was performed on LRRC8A homomers as that is the only available structure, in cells most (if not all) LRRC8-containing channels are heteromeric and contain B-E subunits. If there are good data on LRRC8 subunit expression in BMDMs it would be informative to discuss the likely subunit composition of channels in these cells. Can the authors speculate on whether any of the small molecules would interact less well with heteromeric channels from the docking and sequence differences between LRRC8 subunits? Could this also explain the discrepancy between apparent affinity in HeLa cells and effect on BMDMs for some compounds?

5) Please change "crystal structure" to cryo-EM structure" in the legend for Figure 1.

6) The experiments in Figure 3D-F are a fantastic control that brings clarity to the confusing observation that Cl^-^ efflux is sometimes required for NLRP3 inflammasome activation. It shows that only stimuli that rely on potassium efflux (ATP, Nigericin) are sensitive to Cl^-^channel inhibitors whereas R837, which activates NLRP3 independently of potassium efflux cannot be inhibited by Cl^-^channel inhibitors. This suggests that the Cl^-^ efflux might be linked to K^+^ efflux. For example, it could allow efficient electro-neutral flux. The authors are encouraged to discuss this important experiment and its interpretation more prominently.

[Editors' note: further revisions were suggested prior to acceptance, as described below.]

Thank you for submitting your article "LRRC8A is essential for hypotonicity-, but not for DAMP-induced NLRP3 inflammasome activation" for consideration by *eLife*. Your article has been reviewed by Tadatsugu Taniguchi as the Senior Editor, a Reviewing Editor, and three reviewers. The following individuals involved in review of your submission have agreed to reveal their identity: Murali Prakriya (Reviewer #1); Stephen Graf Brohawn (Reviewer #2).

The reviewers have discussed the reviews with one another and all three are supportive of publication. It was noted that the electrophysiology experiments that were suggested were not completed; however, the reviewers felt that on balance these experiments were not completely essential for the main conclusions of the manuscript.

Overall, we are satisfied that the manuscript represents a significant addition to the literature that clarifies the role of chloride channels, and LRRC8A in particular, in NLRP3 inflammasome activation. We thank you for submitting your outstanding work to *eLife*.

Reviewer #1:

I have gone through the authors responses and the revised manuscript. The authors have revised the emphasis of the paper including the title and the Discussion section sufficiently to address my previous concerns. It would have been nice to include some electrophysiology on LRRC8A but as this is not a major point of the paper, I am satisfied with the revision of the results. I have no further comments or concerns.

Reviewer #2:

I am satisfied with the replies and additions made in response to reviewer comments. I have no further concerns that need to be addressed.

Reviewer #3:

I think the authors have adequately addressed most of the reviewer's comments and I'd recommend the manuscript for publication.

However, there is one point that is still of concern that was not properly addressed by the authors (bullet point 4 in the previous consolidated review). It seems that the authors agree that potassium efflux is the driving force of hypotonicity-induced NLRP3 activation because blocking potassium efflux prevents NLRP3 activation (PMID:22981536). The question is how this relates to VRAC-mediated Cl flux. I think it's an appealing hypothesis that Cl flux needs to happen to allow potassium efflux while maintaining electro-neutrality. The authors make the point that during ATP/Nigericin stimulation this might not be the case, but I'd like to argue that these two situations are likely to differ. It's conceivable that Cl efflux is required for potassium efflux during RVD while Cl efflux is not required for potassium efflux by Nigericin (potassium/proton antiporter) and P2X7 (cation-channel).

Unless the authors provide experimental evidence against the interpretation of their data that Cl efflux during RVD is required to allow potassium efflux, I'd recommend to the authors to mention this hypothesis in the Discussion section.

---

## [Author Response]

Points requiring additional experimentation:1) The pharmacological analysis of VRAC inhibition is done entirely in Hela cells whereas the rest of the paper is about NLRP3 activation in macrophages.a) Given that subunit composition could be different between HeLa cells and macrophages and this could play a role in differing pharmacology, it is imperative that the analysis of VRAC inhibition should be performed on macrophages, not the endogenous channels in Hela cells.

Figure 2C and D are data from primary bone marrow derived macrophages (BMDMs). In this experiment we show that the inhibitors (except for 4-sulfonic[6]calixarene) inhibit the regulated volume decrease (RVD) caused by hypotonic buffer. The original papers characterising the effect of LRRC8A knockdown on VRAC function showed that it was essential for both iodide quenching of YFP fluorescence and for RVD (PMIDs: 24725410, 24790029). In Figure 2 all inhibitors except calixarene blocked RVD in both cell types. Calixarene was identified as a potential VRAC inhibitor from the modelling presented in Figure 1. Thus, the assays in either respective cell type reflect the same effect.

b) In addition, the authors should confirm that the VRAC KOs are completely deficient in RVD by using their VRAC inhibitors in VRAC KOs. In other words Figure 4C should include several of the VRAC inhibitors in both WT and VRAC KO cells.

We have conducted the experiment as suggested by the reviewers. We measured the RVD response in BMDMs from WT and VRAC KO cells in the presence or absence of the established VRAC inhibitor DCPIB. These data show that WT cells treated with the VRAC inhibitor exhibit a similar RVD response to vehicle treated KO BMDMs. Interestingly KO BMDMs appear to exhibit a further swelling response in presence of DCPIB suggesting that in the absence of VRAC other Cl^-^ channels act to constrain cell volume when an RVD is inhibited. These data suggest that DCPIB also inhibits other Cl^-^ channels consistent with our overall observation that VRAC inhibitors block DAMP-induced NLRP3 inflammasome activation by channels other than those dependent on LRRC8A. These data are now provided as Figure 4—figure supplement 1. The text of the manuscript has been amended to reflect these new findings.

2) The cell culture experiments showing hypotonicty-induced NLRP3 activation are convincing, but raise the question: under what circumstance(s) is this physiologically relevant?

Hypo-osmolarity has not been studied extensively in in vivo disease models, but treatment with hypertonic solutions is known to be effective at alleviating inflammation, including in LPS-induced sepsis (PMID: 23295602), intracerebral haemorrhage (PMID: 29700692), pulmonary injury due to ischaemia-reperfusion (PMID: 12511143), and acute respiratory distress syndrome (ARDS) (PMID: 25962375). The reasons for the therapeutic effects of hypertonic fluid treatment are poorly defined, but could be at least in part due to NLRP3 inhibition. Other pathological conditions that cause cell swelling including hypoxia/ischaemia, hyponatremia, hypothermia, and others (PMID: 19126758). Furthermore, animal cell membranes are highly permeable to water and in PMID: 22981536, cell volume regulation is described as a primitive response to alterations in environmental osmolarity. Under physiological conditions cell volume is perturbed by transepithelial transport, accumulation of nutrients and metabolic waste products, and intestinal epithelial cells and blood in intestinal capillaries can be exposed to low extracellular osmolarity after water or hypotonic food intake (reviewed in PMID: 19126758). We have modified the text of the discussion to highlight the potential of hypotonicity induced signalling to disease based on the treatment by hypertonic solutions.

a) There seems to be an underlying implication or assumption that DAMP stimulation evokes VRAC activation. Is this the intendend implication? If so, this needs to be explicitly tested.

Within the literature there was an acceptance that DAMP stimulation could cause changes in cell volume. For example, in 1996 PMID: 8939922 first reported hypotonicity induced IL-1β release, and also showed that ATP (via P2X7 activation)-induced IL-1β release could be inhibited by a hypertonic buffer. In his 2012 paper Pelegrin showed that hypertonic buffer inhibited NLRP3 activation to a range of DAMPs and to hypotonicity. Thus, we assumed that DAMP stimulation would regulate NLRP3 activation via VRAC. We show clearly that this was not the case. This rationale is now clearly explained within the final paragraph of the Introduction.

b) Alternatively, the authors should provide another physiologically relevant scenario in which hypotonicity might naturally be encountered by a cell, leading to NLRP3 activation. Do the authors think that the extreme (though typical in the literature) hypotonic conditions used in culture experiments (using a switch from ~340to ~120 mOsm solutions) are likely to occur in vivo? Under what circumstances?

Please see response to point 2 above.

c) VRAC and LRRC8-contianing channels have been proposed to be activated by stimuli other than reduced intracellular ionic strength. Could these alternatives be involved in LRRC8-mediated NLRP3 activation?

Indeed, for example sphongosine-1-phosphate (S1P) activates VRAC in mouse macrophages (PMID: 24965069). We previously reported that sphingosine, and S1P, could activate NLRP3 (PMID: 22105559). VRAC is also thought to be important for cell death induced by apoptosis inducing drugs (e.g. PMIDs: 26984736, 26530471), and we also previously reported that apoptosis inducing drugs could activate the NLRP3 inflammasome in activated macrophages (PMID: 24790078) suggesting another area of relevance. This has been added to the discussion. LRRC8A was also recently shown to have an important role in anti-viral signalling in response to HSV-1 infection alone by allowing the import and export of the STING activator 2’-3’-cGAMP (PMID: 32277911), suggesting VRAC is activated in these conditions. Several viruses have been speculated to promote activation of the NLRP3 inflammasome (Reviewed by PMID: 30209070), including HSV-1 (PMID: 32101570, 23427152).

d) Finally, can the authors speculate on whether it is Cl^-^ flux per se through LRRC8 or other Cl^-^ channels that is important for NLRP3 activation? Other possibilities include voltage changes, RVD, and flux of other permeant ions like ATP or cyclic dinucleotides.

VRAC is directly permeable to Cl^-^ and we previously provided some mechanistic insight into direct effects of Cl^-^ efflux on NLRP3 dependent ASC oligomerisation (PMID: 30232264). VRAC activation caused by swelling can also activate other Cl^-^ channels, notably anoctamin 1 (e.g. PMIDs: 31242393, 27514381), and we cannot rule out the possibility that other VRAC activated Cl^-^ conductance is involved. Moreover, other studies have also implicated a role for Cl^-^ conductance in NLRP3 activation (PMID 28576828, 28779175, 32523112). VRAC is also permeable to some small molecules, and recently VRAC was reported to act as a conduit for cGAMP and contribute to antiviral immune responses in bystander cells as a result (PMID: 32277911). VRAC can also regulate the release of ATP (PMID: 31988241) which could activate P2X7, although hypotonicity induced NLRP3 activation was established to be independent of P2X7 (PMID: 22981536). The Discussion section has been modified to include the reviewers point.

e) In addition to the above (or as a result of addressing the above), the authors should expand on the discussion of potential physiological implications.

Please see point 2 above.

3) The analysis of VRAC channel function based solely on the YFP-quenching assay is technically inadequate. The differential effects on NLRP3 activation of the different inhibitors is a potentially important finding, but this claim cannot be made solely on this one indirect readout. The YFP quenching method is unlikely to have sufficient temporal resolution and dynamic range to detect smaller changes (or lack of changes) in ionic currents. Direct electrophysiological analysis of the chloride currents is required to test if there are differences in the mechanism or time course of inhibition that could explain the lack of effects on DAMP mediated inflammasome activation.

In Figure 2 we present data on RVD alongside the hypotonicity induced quenching, which are both dependent upon LRRC8A/VRAC (PMIDs: 24725410, 24790029). Many of the inhibitors used have been tested against VRAC using electrophysiology and are known to be effective e.g. tamoxifen and NS3728 (PMID: 14724745), fenamates (PMID: 27509875), DCPIB (PMID: 28620305), and MONNA was shown to be effective in the quenching assay described here (PMID: 27219012), although whether these are altered by DAMP stimulation is unknown. In any case as we show that VRAC is dispensible for DAMP-induced NLRP3 activation it is less critical to the point of the paper.

4) The authors should investigate if VRAC-mediated Cl^-^efflux enables K^+^ efflux to drive NLRP3 inflammasome activation during hypo-osmolar conditions. They should determine the intracellular K^+^ and Cl^-^ content in presence of pharmacological NLRP3 inhibition in WT vs VRAC KO cells. I think it'll be important to correlate the efflux of K^+^ with NLRP3 activation during RVD, especially in light of PMID: 22981536, which claims otherwise (using pharmacological inhibition).

Using ion substitution experiments we were able dissociate the effects of Cl^-^ and K^+^ efflux which regulate different aspects of the NLRP3 activation process (PMID: 30232264). However, it is clear that Cl^-^ changes go hand in hand with K^+^ efflux to coordinate NLRP3 activation. This was also shown by PMID:22981536 (same paper as suggested by reviewer) who showed that high extracellular K^+^ inhibited hypotonicity induced NLRP3 activation. We and others have previously shown that Cl^-^inhibitors are ineffective at blocking K^+^ efflux to DAMPs (PMID: 30232264, 29021150) and in the case of ATP and nigericin-induced NLRP3 activation, knockdown of several CLICs reduced Cl^-^ efflux without altering K^+^ efflux (PMID: 28779175).

Points requiring textual revisions to the manuscript:1) The title should be reworked to include the key finding that LRRC8a does not regulate canonical NLRP3 inflammsome activation as this is a central part of the paper. The title only reflects Figure 6 while most of the work shows that LRRC8a does not regulate the canonical NLRP3 inflammasome.

We have changed the title to: ‘LRRC8A is essential for hypotonicity-, but not for DAMP-induced NLRP3 inflammasome activation’

2) There is an underlying assumption that the inhibitors that inhibit canonical (ATP for instance) NLRP3 activation (MON, FFA, NS3) likely inhibit another Cl^-^ channel. This is part of the discussion and interpretation of their data and it appears there is no rational basis for this conclusion. The mechanism for how these compounds block NLRP3 remains a mystery and could very likely include other unknown targets.

This is true, whilst there is evidence supporting the importance of Cl^-^ in the activation of NLRP3 in response to DAMPs, and that Cl^-^ channel inhibitors are promiscuous for other Cl^-^ channels, there is the possibility that the inhibitors are hitting a different target. This caveat has been added to the discussion.

3) The authors are suggested to modify the presentation of Figure 1 for clarity.a) In (A), two orthogonal views (a side view with one or two LRRC8A subunits in the front removed and a top view) and the addition of lines indicating membrane boundaries would give better perspective.

An additional view has been added and membrane boundaries indicated.

b) The legend states B-I are colored with "hydrophilic (blue), hydrophobic regions (red) and neutral regions (white)." It is recommended to change the coloring in B-I are colored to the more typical blue (electropositive) to white (electroneutral) to red (electronegative) as in Figure 1—figure supplement 1. Or to a green-purple hydrophobic/hydrophilic color scheme as red/white/blue is typically used for electronegative-electropositive coloring. In either case, please add a key to the figure with the color scale.

Electronegative (red), electropositive (blue) and electroneutral (white) colouring has now been used to make this the same as in Figure 1—figure supplement 1. A key has been added to Figure 1 to reflect this and the legend has been modified.

c) Consider adding labels for each drug in the panels.

Drug names have been added to each panel in Figure 1.

4) While the docking was performed on LRRC8A homomers as that is the only available structure, in cells most (if not all) LRRC8-containing channels are heteromeric and contain B-E subunits. If there are good data on LRRC8 subunit expression in BMDMs it would be informative to discuss the likely subunit composition of channels in these cells. Can the authors speculate on whether any of the small molecules would interact less well with heteromeric channels from the docking and sequence differences between LRRC8 subunits? Could this also explain the discrepancy between apparent affinity in HeLa cells and effect on BMDMs for some compounds?

We don’t present any discrepancy between HeLa and BMDM cells. The data presented in Figure 2 are entirely consistent with each other. From a RNAseq analysis of a previous study (PMID: 30444603) we find LRRC8D the most abundantly expressed form in BMDMs. Recent work on the cryo-EM structure of the LRRC8D homohexamer to have a wider pore potentially allowing greater substrate permeation than the purely LRRC8A form (PMID: 32415200). However, the physiologically relevant heterohexameric complexes in these cells remain to be determined, as does whether multiple channels of different sub-unit composition can be differentially functionally active in the same cell. This is an area for further research.

5) Please change "crystal structure" to cryo-EM structure" in the legend for Figure 1.

Thank you, this has now been corrected.

6) The experiments in Figure 3D-F are a fantastic control that brings clarity to the confusing observation that Cl^-^ efflux is sometimes required for NLRP3 inflammasome activation. It shows that only stimuli that rely on potassium efflux (ATP, Nigericin) are sensitive to Cl^-^channel inhibitors whereas R837, which activates NLRP3 independently of potassium efflux cannot be inhibited by Cl^-^channel inhibitors. This suggests that the Cl^-^ efflux might be linked to K^+^ efflux. For example, it could allow efficient electro-neutral flux. The authors are encouraged to discuss this important experiment and its interpretation more prominently.

As discussed above, using ion substitution experiments we were able dissociate the effects of Cl^-^ and K^+^ efflux which regulate different aspects of the NLRP3 activation process (PMID: 30232264). The experiments here in Figure 3D-F show that Cl^-^ channel inhibition selectively inhibits NLRP3 inflammasomes that require K^+^ efflux. We agree that this is a very exciting finding and will form a significant component of our ongoing research. Such a discovery allows the selective therapeutic targeting of canonical NLRP3 inflammasomes dependent on K^+^ efflux using Cl^-^ channel inhibitors. This compliments other approaches using direct NLRP3 inhibitors and may allow targeted inhibition of NLRP3 whilst minimising immunosuppression. The importance of this observation has now been further emphasised in the Discussion section.